# Unleashing the Information Flow: Graph Neural Networks are Noisy Communication Channels

## Abstract

Existing message-passing graph neural networks often rely on carefully designed information propagation methods to perform reasonably in graph-related mining tasks. However, this invokes the problem of whether the dimensions of learnable matrices and the depths of the networks are properly estimated. While this challenge has been attempted by others, it remains an open problem. Using the principle of maximum entropy and Shannon's theorem, we demonstrate that message-passing graph neural networks function similarly to noisy communication channels. The optimal information transmission state of graph neural networks can be reached when Shannon's theorem is satisfied, which is determined by their entropy and channel capacity. In addition, we illustrate that the widths of trainable matrices should be sufficiently large to avoid the shrinkage of model channel capacity and the increase of the channel capacity diminishes as the depth of the networks increases. The proposed approach is empirically verified through extensive experiments on five public semi-supervised node classification datasets.

## 1 Introduction

The message-passing graph Neural Networks (GNNs) have been favored choices in various graph-based tasks, which arises from their powerful ability to exploit the connectivity information and node attribute features to realize effective information propagation over the graphs. For instance, GCN (Kipf & Welling, 2017) defines the information propagation operation based on the first-order approximation of the spectral filter on graphs, GAT (Veličković et al., 2018) extends this operation by adaptively assigning edge weights through the multi-head graph attention, and GDC (Gasteiger et al., 2019) further relaxes this by redefining adjacency matrices with kernel functions (e.g., heat kernel (Chung, 2007) and Personalized PageRank (Wang et al., 2020)). Sharing the core idea of propagating graph features with adjacency information, many powerful GNNs have been proposed such as APPNP (Gasteiger et al., 2018), SGC (Wu et al., 2019), GATv2 (Brody et al., 2022), etc. Nonetheless, the depths and widths of most message-passing GNNs are selected based on heuristical guidelines and non-trivial engineering knowledge (Cai et al., 2021; Xu et al., 2023).

Though the connectivity information plays a crucial part in information exchange over graphs, the dimensionalities of learnable matrices and the depth of the network are inescapable aspects to be considered (Chen et al., 2020b; Zeng et al., 2021). Several studies (Liu et al., 2020; Li et al., 2021) show that deep GNNs can benefit from larger receptive fields, while others (Wu et al., 2019; Eliasof et al., 2021) debate that shallow GNNs can avoid critical problems like over-smoothing in deep GNNs. From the view of signal processing, learnable matrices can be seen as the encoders of latent domains, of which their dimensionalities explicitly determine the transformation of graph signals between domains. Hence, some studies have explored the potential effects of GNN's depths and widths. For instance, GraphGym (You et al., 2020) provides a general search framework for selecting the best GNN from a pre-defined search space for a specific downstream task. It comprises multiple aspects of constructing GNNs, such as the network depth, the widths of trainable matrices, the aggregation method, and the activation function. This work facilitates the development of GNNs for various graph-mining tasks from an empirical aspect. Despite comprehensive empirical guidelines being concluded based on extensive experiments, mechanisms behind certain configurations of depths and widths are not yet demystified. While Loukas (2020) argues that existing GNNs

often fail in effectively passing information over the graph unless certain products of their depths and widths exceed a polynomial of graphs. This work illustrates that the expressive ability of GNNs depends on their model capacity, which is directly determined by the widths and depths of the networks. More recently, another work, MGEDE (Yang et al., 2023) has attempted to bridge this gap by introducing the minimum entropy principle to learn a highly informative yet low-dimensional representation. First, it defines the entropy of a graph by a direct summation of its structural entropy and node attribute entropy. Subsequently, it relates the layer widths to the graph entropy based on the distributional hypothesis (Sahlgren, 2008) with several numerical approximations. Eventually, the layer widths are determined by zeroing out the graph entropy according to the minimum entropy principle. Nonetheless, MGEDE neither provides justifications nor explanations about the chosen depths for message-passing. More importantly, the minimum entropy principle, or Prigogine's theorem (Prigogine, 1978) is narrowly valid in certain near-equilibrium linear systems, which might not be suitable for nonlinear systems like GNNs.

In the face of properly estimating the depths and widths of GNNs in advance, we propose a novel theoretical approach termed Channel Capacity Constrained Estimation (C3E). First, we define the entropy and the channel capacity for GNNs. Then, unlike conventional ways to select the widths and depths based on empirical knowledge and grid search, C3E obtains feasible widths and depths of a GNN by solving a nonlinear mathematical programming problem given the input graph. Eventually, based on the feasible solutions, we empirically demonstrate that GNNs are governed by their entropy and channel capacity.

## 2 OTHER RELATED WORK

### 2.1 INFORMATION THEORY IN DEEP LEARNING

Information theory (Shannon, 1948; Jaynes, 1957; Kullback, 1997) has long been a powerful tool for analyzing complex nonlinear systems like neural networks. For instance, some studies (Chan et al., 2022; Roberts et al., 2022) endeavor to establish statistical relationships between latent representation and neural networks. The information bottleneck theory is broadly adopted in neural networks to learn minimal effective representations that maximize the mutual information between the learned representation and the output (Tishby et al., 2000; Wu et al., 2020). Similarly, another work (Yu et al., 2020) proposes learning more discriminative and diverse representations via maximal coding rate reduction. In the meantime, Saxe et al. (2019) have explored the entropy distribution and the information flow of deep neural networks. Furthermore, Sun et al. (2021) and Shen et al. (2023) have attempted to utilize the principle of maximum entropy to link latent feature maps to the distinctive behaviors of neural networks.

### 2.2 EXPLORATION IN NEURAL NETWORK ARCHITECTURE

Early works (e.g., Wang & Zhu, 2021; Cai et al., 2021; Zhou et al., 2022) adopt computationally intensive methods like automated machine learning and network architecture search to generate neural networks. The research focus has then shifted towards principled theoretical analysis of the behaviors governing neural networks. For example, DeepMAD (Shen et al., 2023) established the statistical relationship between architectures of convolutional neural networks (CNNs) and Shannon's entropy; Di Giovanni et al. (2023) have investigated the potential impacts of GNNs' depths and widths from over-squashing and over-smoothing; GCNII (Chen et al., 2020a) empirically demonstrates that skip connections and identity mapping can improve GNNs' performance and stack deeper, and AERO-GNN (Lee et al., 2023) realizes deeper attention-based GNNs by modifying the attention mechanism and introducing weighted skip connections.

## 3 PRELIMINARY

**Entropy of Random Variable**. We can define the entropy of a matrix $\boldsymbol{Z} \in \mathbb{R}^{\alpha \times \beta}$ by regarding it as a matrix-valued continuous random variable $\mathbf{Z} = \{\boldsymbol{Z}^{i,j} \mid i \leq \alpha, j \leq \beta\}$,

$$H(\mathbf{Z}) = -\int p(z) \log{(p(z))} dz. \tag{1}$$

Furthermore, this real-valued random variable reaches its maximum entropy by the corresponding Gaussian distribution $\mathcal{N}(\mu(\mathbf{Z}), \sigma^2(\mathbf{Z}))$,

$$H(\mathbf{Z}) \leq \frac{1}{2} \log\left(2\pi e \sigma^2(\mathbf{Z})\right). \tag{2}$$

The proof is provided in the Appendix A.1. However, its probability distribution $p(z)$ is often unknown in many real-world applications. In practice, we can adopt its discrete form as a direct approximation,

$$H(\mathbf{Z}) = -\sum_{z \in \mathbf{Z}} P(\mathbf{Z} = z) \log\left(P(\mathbf{Z} = z)\right). \tag{3}$$

**Entropy of Graphs**. The graph entropy provides a quantitative measure of its structural complexity and information content. Though definitions of graph entropy vary across different areas of research, they share the same underlying principle of quantifying potential states of nodes in the discrete form. The graph entropy can be generalized as the following form (Dehmer & Mowshowitz, 2011)[Def. 2.12],

$$H(\mathcal{G}) = -\sum_{i=1}^{n} \frac{g(v_i)}{\sum_{i=1}^{n} g(v_i)} \log \frac{g(v_i)}{\sum_{i=1}^{n} g(v_i)}. \tag{4}$$

Here, $g(\cdot)$ denotes the information extraction function, $\mathcal{G}$ denotes an arbitrary finite graph, $v_i$ denotes $i$-th vertex, and $n$ denotes the number of nodes. The calculation of the probability distribution takes a similar form to Boltzmann distribution (Gibbs, 1902; Boltzmann, 2015), thereby $H_{\max}(\mathcal{G}) = \log n$.

**Channel Capacity.** According to Shannon (1948), the channel capacity of a noisy communication channel is expressed as,

$$\phi = \max_{p_{\mathbf{Q}}(q)} I(\mathbf{Q}; \mathbf{Q}'). \tag{5}$$

Here, $I(\mathbf{Q}; \mathbf{Q}')$ denotes the mutual information between the input $\mathbf{Q}$ and the output $\mathbf{Q}'$, and $p_{\mathbf{Q}}(q)$ denotes the corresponding marginal distribution. In terms of information theory (Shannon, 1948; Jaynes, 1957; Gallager, 1968), channel capacity is defined as the theoretical maximum rate at which information can be reliably transmitted over a noisy communication channel with an arbitrarily small error rate. Concretely, Shannon's theorem (Shannon, 1948) states that given a noisy channel with channel capacity $\phi$ and information rate $H$, there exists a coding scheme that allows the probability of error at the receiver to be made arbitrarily small when,

$$\phi \geq H. \tag{6}$$

## 4 ENTROPY OF GRAPH NEURAL NETWORK

Suppose a GNN with $L$ message-passing layers (denoted as $\Omega_L$) performing a semi-supervised node classification task on an arbitrary finite graph characterized by a node feature matrix $\boldsymbol{X} \in \mathbb{R}^{n \times m}$ and an adjacency matrix $\boldsymbol{A} \in \mathbb{R}^{n \times n}$. Then, its layer propagation rule can be recursively expressed as,

$$\boldsymbol{H}_l = \gamma(\tilde{\boldsymbol{C}}_l \boldsymbol{H}_{l-1} \boldsymbol{W}_l). \tag{7}$$

Here, $\boldsymbol{H}_l \in \mathbb{R}^{n \times w_l}$ denotes the latent node representation, $\boldsymbol{W}_l \in \mathbb{R}^{w_{l-1} \times w_l}$ denotes the trainable weight matrix, $\tilde{\boldsymbol{C}}_l \in \mathbb{R}^{n \times n}$ denotes any generalized form of adjacency matrix by either functions or neural networks, and $\gamma(\cdot)$ denotes the activation function. Let's consider the most common scenario $\tilde{\boldsymbol{C}}_l = \tilde{\boldsymbol{C}}_{l-1} = \cdots = \tilde{\boldsymbol{C}}_1 = \tilde{\boldsymbol{A}} = \boldsymbol{D}^{-\frac{1}{2}} \boldsymbol{A} \boldsymbol{D}^{-\frac{1}{2}}$, where $\boldsymbol{D} \in \mathbb{R}^{n \times n}$ denotes the node degree matrix, and $d \in \mathbb{R}^+$ denotes the average (node) degree.

On the one hand, from a statistical physics view, the principle of maximum entropy suggests that the equilibrium state of a closed system should be taken in the state where the entropy of the system is at its maximum under the given constraints (Callen, 1991). On the other hand, from a signal-processing perspective, the principle of maximum entropy states that given a set of known constraints, the probability distribution that best represents the current state of knowledge about an unknown system is the one with the maximum entropy (Jaynes, 1957). If such a maximum entropy state of a physical system corresponds to the optimal transmission state in an information processing system (Jaynes, 1957; Von Neumann, 2018), then the entropy of such an information processing system in this state corresponds to the upper bound of its entropy.

In real-world applications, the node representation has finite dimensions, and its distribution is unknown in advance. Its maximum entropy can be approximated in the discrete form. In the meantime, $\gamma(\cdot)$ is applied element-wise, which does not influence dimensionalities of $\boldsymbol{H}_l$ (i.e., whatever the activation function is, $n$ and $w_l$ do not change). Accordingly, the maximum entropy of $\boldsymbol{H}_l$ remains unchanged, and note that though its maximum entropy is unchanged, its entropy is changed due to the distribution shifts incurred by the activation function. Therefore, Eq. (7) can be further simplified as,

$$\boldsymbol{H}_l = \boldsymbol{U}_l \boldsymbol{W}_l$$
$$= (\tilde{\boldsymbol{A}} \boldsymbol{H}_{l-1}) \boldsymbol{W}_l$$
$$= \tilde{\boldsymbol{A}} \left( \tilde{\boldsymbol{A}} \left( \cdots \left( \tilde{\boldsymbol{A}} \left( \tilde{\boldsymbol{A}} \boldsymbol{H}_0 \boldsymbol{W}_1 \right) \boldsymbol{W}_2 \right) \cdots \right) \boldsymbol{W}_{l-1} \right) \boldsymbol{W}_l. \tag{8}$$

As mentioned previously, the principle of maximum entropy states that the probability distribution that best represents the current state of a system is the one with the largest entropy. This leads us to the following Theorem,

**Theorem 1.** *Assume* $\mathbf{W}_l \sim \mathcal{N}(0,1)$, $\mathbf{H}_0 \sim \mathcal{N}(0,1)$, *then the mathematical expectation of* $\mathbf{H}_l$ *is expressed as,*

$$\mathbb{E}(\mathbf{H}_l) = w_{l-1} \mathbb{E}(\mathbf{W}_l) \mathbb{E}(\mathbf{U}_l) = 0. \tag{9}$$

*and the variance of* $\mathbf{H}_l$ *is expressed as,*

$$\sigma^2(\mathbf{H}_l) = \prod_{o=0}^{l-1} n w_o \sigma^2(\tilde{\boldsymbol{A}}) \approx \prod_{o=0}^{l-1} \frac{w_o}{d}. \tag{10}$$

*Since the ideal GNN is not observable in advance, the GNN with the maximum entropy should be the choice. Therefore, the GNN in its optimal transmission state is reached by maximizing its entropy, which is defined by,*

$$H(\Omega_L) = \frac{1}{2} \log(2\pi e) + \frac{1}{2} \sum_{l=0}^{L-1} \log(w_l) + \frac{1}{2} \sum_{l=0}^{L-1} \log\left(n\sigma^2(\tilde{\mathbf{A}}_l)\right)$$

$$\approx \frac{1}{2} \log(2\pi e) + \frac{1}{2} \log(w_L) + \sum_{l=0}^{L-1} \frac{1}{2} \log\left(\frac{w_l}{d}\right). \tag{11}$$

The derivations are provided in the Appendix A.2. We can conclude the following two points based on the above Theorem and empirical results. First, if the latent node representation is properly regularized or normalized, then $\mathbb{E}(\mathbf{H}_l)$ approaches zero, which implies no significant bias terms present. Second, $\sigma^2(\mathbf{H}_l) = \prod_{o=0}^{l-1} n w_o \sigma^2(\tilde{\boldsymbol{A}}) \approx \prod_{o=0}^{l-1} \frac{w_o}{d}$, this term explicitly captures the essence of the expressiveness of GNNs or the diversity of latent node representation. If $\tilde{\boldsymbol{A}}_l$ is invariant and $w_l$ is not sufficiently large compared to $d$, then $\sigma^2(\mathbf{H}_l)$ approaches zero as the network stacks deeper. This can result in a negative entropy of the GNN, which suggests a high degree of concentration and a small spread out in the distribution it represents. In other words, the values of latent features cluster around a certain point, making the feature patterns overly similar and highly biased. While this paper originally intends to present a principled theoretical framework to understand the different behaviors of GNNs, it offers a viewpoint on the inherent drawbacks of GNNs such as over-smoothing.

## 5 CHANNEL CAPACITY OF GRAPH NEURAL NETWORK

However, maximizing Eq. (11) does not guarantee either the performance or the convergence of the GNN. The number of learnable parameters is at least $\sum_{l=1}^{L} w_l w_{l-1}$, which can result in an overly deep and complex neural network when multiple layers are stacked. The deep and complex neural networks typically lead to exploding or vanishing gradients, causing weight updates to be drastic and diverge in training. Although not concentrating on GNNs, previous works (Roberts et al., 2022; Shen et al., 2023) introduce a metric termed aspect ratio $\rho$ to control the information propagated in deep neural networks,

$$\rho = \frac{L}{\bar{w}}, \bar{w} = \left(\prod_{l=1}^{L} w_l\right)^{\frac{1}{L}}. \tag{12}$$

According to Roberts et al. (2022), the neural networks, on the one hand, become simple linear models if $\rho \to 0$. On the other hand, the neural networks perform like chaos systems if $\rho \to \infty$. Nonetheless, the determination method of this metric for specific neural networks is not provided while Shen et al. (2023) treats it as a tunable hyperparameter.

Recall that the channel capacity is defined as the theoretical maximum of the mutual information between the input and the output of the channel. Therefore, we can alter the channel capacity of an information processing system to control the effective information transmitted throughout. Ideally, if there exists a GNN that learns the exact mapping between input graph $\mathcal{G}$ and output graph $\mathcal{G}'$, then,

$$H(\mathcal{G}'|\Omega_L(\mathcal{G})) = H(\Omega_L(\mathcal{G})|\mathcal{G}') = 0. \tag{13}$$

Based on Eq. (5) and Eq. (13), the channel capacity of an $L$-layer GNN is defined by,

$$\phi(\Omega_L) = H(\Omega_L(\mathcal{G})) = H(\mathbf{H}_L). \tag{14}$$

However, an observer can not know or precisely estimate $\mathcal{G}'$ prior, otherwise neural networks are not required to learn the latent mapping relationship. Fortunately, Shannon's Theorem still holds if the channel capacity of a GNN is not less than the maximum entropy of $\mathcal{G}'$,

$$\phi(\Omega_L) \geq H_{\max}(\mathcal{G}'). \tag{15}$$

In general, if a GNN achieves the optimal information transmission state, its lower bound should be no less than the maximum entropy of $\mathcal{G}'$, which can be summarized as the following Theorem.

**Theorem 2.** *If the lower bound $\phi_0$ satisfies $\phi(\Omega_L) \geq \phi_0 \geq H_{max}(\mathcal{G}')$, then $\phi(\Omega_L) \geq H_{max}(\mathcal{G}') \geq H(\mathcal{G}')$. Consequently, the information transmission with an arbitrarily small error rate between $\mathcal{G}$ and $\mathcal{G}'$ can be reached, such that $\phi_0 = \sum_{l=1}^{L} \frac{1}{l} \log\left(\frac{w_{l-1}w_l}{w_{l-1}+w_l}\right)$.*

In fact, real-world communication channels suffer from internal interference and signal attenuation, significantly reducing available channel capacity. These could correspond to analogous phenomena that occur in neural networks, such as gradient interference (Xu et al., 2022) and vanishing gradients and exploding gradients (He et al., 2016), leading to a considerable reduction in effectively propagating information. Hence, an enforced lower bound of $\phi(\Omega_L)$ is essential to guarantee the requirements in Eq. (15), which is expressed as,

$$\begin{aligned}
\phi(\Omega_L) &\geq H(\mathbf{H}_L) - H(\mathbf{H}_0) \\
&\geq \sum_{l=1}^{L} \frac{H(\mathbf{H}_l) - H(\mathbf{H}_{l-1})}{H(\mathbf{H}_{l-1})} H(\mathbf{H}_{l-1}) \\
&\geq \sum_{l=1}^{L} \frac{H(\mathbf{H}_l) - H(\mathbf{H}_{l-1})}{H(\mathbf{H}_{l-1})} I(\mathbf{H}_l; \mathbf{H}_{l-1}) \\
&\geq \sum_{l=1}^{L} \frac{1}{l} \log\left(\frac{w_{l-1}w_l}{w_{l-1}+w_l}\right).
\end{aligned} \tag{16}$$

The derivations are provided in the Appendix A.3. This lower bound shows that the channel capacity of each layer is dependent on the harmonic mean of two adjacent layers, scaled by the inverse of its depth. It indicates that the increase of the channel capacity decreases drastically when solely increasing the depth, such that its negative effects always surpass its merits (typically reflected as model performance degradation).

## 6 CONSTRAINED NONLINEAR PROGRAMMING PROBLEM

According to Theorem 1 and Theorem 2, estimating the widths and depth of an $L$-layer GNN can be then transformed into solving a nonlinear constrained mathematical programming problem. Previous works (He et al., 2016; Huang et al., 2017; Li et al., 2021) suggest that a neural network refines and compresses information as depth increases. Hence, we can have the following relationship based on Eq. (3),

$$w_1 \geq w_2 \cdots \geq w_{L-1} \geq w_L. \tag{17}$$

Table 1: Semi-supervised node classification results (mean of 10 runs). Best performance is highlighted in bold, while the underlined values denote a statistically significant improvement over other alternatives (Kruskall-Wallis test with Dunn's post-hoc: $p < 0.05$)

| Model | Cora | Citeseer | Pubmed | AmazonPhoto | AmazonComputers |
|-------|------|----------|--------|-------------|-----------------|
| Plain-GCN | 0.8087 | 0.7016 | 0.7852 | 0.8997 | 0.8038 |
| Deep-GCN | 0.7762 | 0.6707 | 0.7519 | 0.8912 | 0.7966 |
| C3E-GCN | **0.8385** | **0.7263** | **0.7982** | **0.9177** | **0.8366** |
| Plain-GAT | 0.8264 | 0.7119 | 0.7858 | 0.9036 | 0.8077 |
| Deep-GAT | 0.8043 | 0.7076 | 0.7607 | 0.9022 | 0.8092 |
| C3E-GAT | **0.8304** | **0.7201** | **0.7899** | **0.9152** | **0.8307** |
| Plain-GDC | **0.8275** | 0.7164 | **0.7956** | 0.8967 | 0.8096 |
| Deep-GDC | 0.7991 | 0.6917 | 0.7679 | 0.9093 | 0.8265 |
| C3E-GDC | 0.8229 | **0.7268** | 0.7890 | **0.9158** | **0.8447** |

Ultimately, the nonlinear programming problem becomes,

$$\max_{\mathbf{w}^{(L)}} \sum_{L=2} \frac{1}{2} \log{(2\pi e)} + \frac{1}{2} \log{(w_L)} + \sum_{l=0}^{L-1} \frac{1}{2} \log{(\frac{w_l}{d})}, \tag{18}$$

$$\text{s.t.} \sum_{l=1}^{L} \frac{1}{l} \log{(\frac{w_{l-1}w_l}{w_{l-1}+w_l})} \geq \log n,$$

$$w_1 \geq w_2 \cdots \geq w_{L-1} \geq w_L.$$

Here, $\mathbf{w}^{(L)} = \{w_1, w_2, \ldots, w_L\}$. The feasible solutions can be obtained by off-the-shelf solvers for constrained nonlinear programming (with inequation constraint available), such as SLSQP (Kraft, 1988). This solving process terminates when $\sum_{l=1}^{L} \frac{1}{l} \log{(\frac{w_{l-1}w_l}{w_{l-1}+w_l})} - \log n > \zeta \log n$.

## 7 EXPERIMENTS

In this work, three mainstream backbone models (GCN (Kipf & Welling, 2017), GAT (Veličković et al., 2018), and GDC (Gasteiger et al., 2019)) on semi-supervised node classification are selected to avoid biased results, as most message-passing GNNs are generally variants of the three models. Here, Plain-XXX refers to the plain backbone model, C3E-XXX refers to the variant generated by C3E, and Deep-XXX refers to the variant with the same width as the original backbone model and the same other configuration as C3E variants. Detailed model configurations and dataset information are provided in Appendix E and C. The code is anonymized and is available at **here**.

### 7.1 PERFORMANCE EVALUATION

Table .1 presents the semi-supervised node classification results of backbone models and their variants. From Table .1, we can observe that C3E variants generally outperform other models, though they underperform Plain-GDC in Pubmed and Cora. Meanwhile, we observe that the Deep variants underperform the plain backbone models and C3E variants in most cases. Furthermore, C3E-GCN has an average performance improvement of 2.34% over Plain-GCN, C3E-GAT has an average performance improvement of 0.64% over Plain-GAT, and C3E-GDC has an average performance improvement of 0.56% over Plain-GDC. These suggest that except for considering the information propagation mechanism, simply stacking more layers without properly estimating layer width often leads to severe performance degradation.

### 7.2 NOISY COMMUNICATION CHANNEL

From Fig. 1, we can observe that $H(\Omega_L)$ monotonically increases as $\phi_0$ increases. Meanwhile, the model with higher entropy tends to perform better given a certain channel capacity. The accuracy

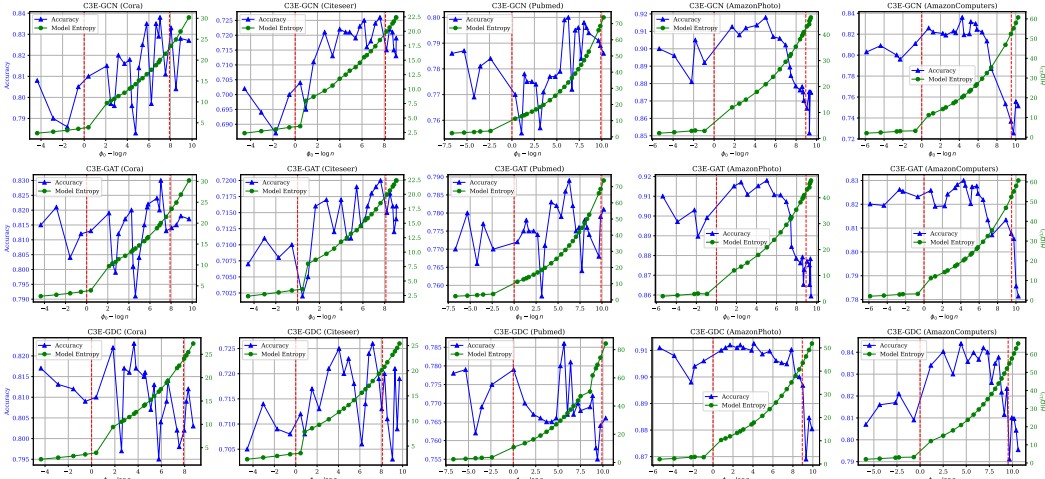

Figure 1: The x-axis denotes $\phi_0 - \log n$, left y-axis denotes classification accuracy, and right y-axis denotes $H(\Omega_L)$. The blue lines denote $\phi_0 - \log n$ versus accuracy, the green lines denote $\phi_0 - \log n$ versus $H(\Omega_L)$, the intervals formed by the red vertical lines represent $[\phi_0 - \log n, \phi_0 - 2\log n]$, and each point represents a C3E model with different combinations of width and depth.

curves exhibit continuous U-shaped and V-shaped patterns over the figures. This somehow coincides with a phenomenon that the model performance displays consistent U-shaped curves as the number of model parameters grows (Hoffmann et al., 2022).

In addition, we observe that C3E variants reach their peak performance within $[\phi_0 - \log n, \phi_0 - 2\log n]$, after and before which the optimal performance cannot be achieved. On the one hand, we observe that the model performance consistently fluctuates before $\phi_0 - \log n$ without a clear trend of either increasing or decreasing. This suggests that GNNs act like band-limited communication channels, where the information transmitted over them is incomplete and noisy. On the other hand, we can observe that when the channel capacity exceeds certain thresholds, the performance of models can degrade drastically. This implies that, unlike real communication channels, GNNs might suffer from severe over-fitting due to extra available channel capacity. In terms of information propagation mechanisms, C3E-GCN, C3E-GAT, and C3E-GDC achieve their maximum performance with dissimilar channel capacities. Moreover, C3E-GDC can reach its peak performance earlier or later than C3E-GCN and C3E-GAT due to its unique mechanism. Unlike the graph attention mechanism adaptively weighs the given edges, the adjacency matrices are transformed and sparsified in GDC models (e.g., unify $d$ to 64 or 128, or zero out entries below certain thresholds), such that the original graphs are reshaped into new graphs. This further results in inconsistent behaviors of C3E-GDC across the datasets, i.e., the rewired graph changes its intrinsic information according to Eq. (10).

## 7.3 ENTROPY TRANSITION OF NODE REPRESENTATION

As demonstrated in Fig. 2, the node representation of GNNs experiences an entropy increase-to-decrease process. The entropy increase indicates that GNNs gradually learn more complex and informative features, while the significant entropy reduction in the last two layers suggests those learned latent features are then converged into more deterministic ones for classification tasks. Moreover, different information propagation mechanisms lead to distinguishable entropy change curves, e.g., GAT-based models are more stable than GCN-based models and GDC-based models. The massive entropy reduction occurs relatively early in most Deep models, implying diverse features are collapsed into centralized or biased features. Unlike other models, C3E variants exhibit a clear entropy transition process with smaller variances. Concretely, C3E variants experience a considerable entropy increase at the first layer, gradually increasing to their maximums in the following layers, and decreasing substantially at the last two layers. Yet, these characteristics do not necessarily guarantee better performance in the downstream tasks as shown in Table .1.

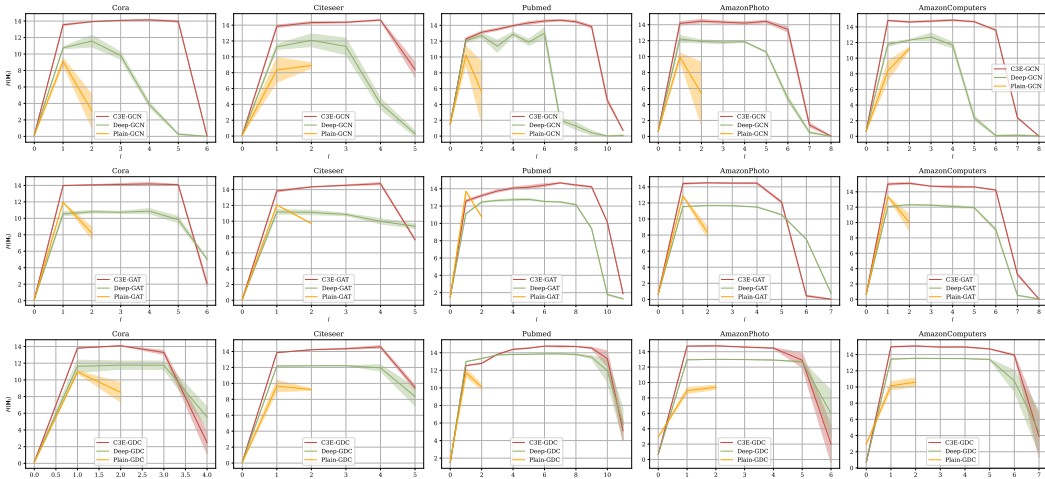

Figure 2: The transitions of $H(\mathbf{H}_l)$ of C3E variants, the Deep variants, and plain backbone models. The shaded area represents the corresponding variance. The entropy of node representation generally exhibits a massive increase in shallow layers and then converges to certain smaller values at the last linear reshaping layer.

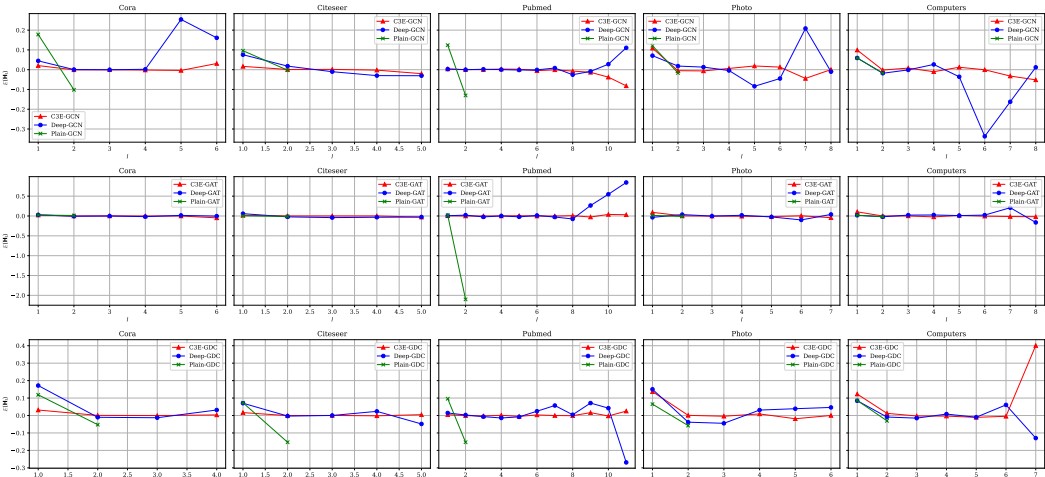

Figure 3: The layer-wise changes of $\mathbb{E}(\mathbf{H}_l)$ of C3E variants, the Deep variants, and plain backbone models. Among the experimented models, $\mathbb{E}(\mathbf{H}_l)$ of C3E variants is much closer to zero than other models, and C3E-GAT has the most stable $\mathbb{E}(\mathbf{H}_l)$ retaining almost zero.

## 7.4 MEAN & VARIANCE OF NODE REPRESENTATION

To verify the proposed theorems, we provide the layer-wise $\mathbb{E}(\mathbf{H}_l)$ and $\sigma^2(\mathbf{H}_l)$ in Fig. 3 and Fig. 4. Since $\mathbb{E}(\mathbf{H}_l)$ and $\sigma^2(\mathbf{H}_l)$ have a close relationship with $H(\mathbf{H}_l)$, we can gain some insights into the internal dynamics of GNNs. Recall that,

$$\mathbb{E}(\mathbf{H}_l) = w_{l-1}\mathbb{E}(\mathbf{W}_l)\mathbb{E}(\mathbf{U}_l) = 0, \ \sigma^2(\mathbf{H}_l) = \prod_{o=0}^{l-1} n w_o \sigma^2(\tilde{\mathbf{A}}) \approx \prod_{o=0}^{l-1} \frac{w_0}{d}. \tag{19}$$

Based on Fig. 3, in general, $\mathbb{E}(\mathbf{H}_l)$ of C3E models and the Deep models are approximately zero except on a few datasets in the last two layers. Furthermore, we can observe that among the experiment models, C3E variants demonstrate the most stable transitions in terms of $\mathbb{E}(\mathbf{H}_l)$ across five datasets, while $\mathbb{E}(\mathbf{H}_l)$ of Deep models exhibit more versatile patterns. This suggests that simply stacking multiple layers or adopting shallow models cannot guarantee alleviating the biased terms

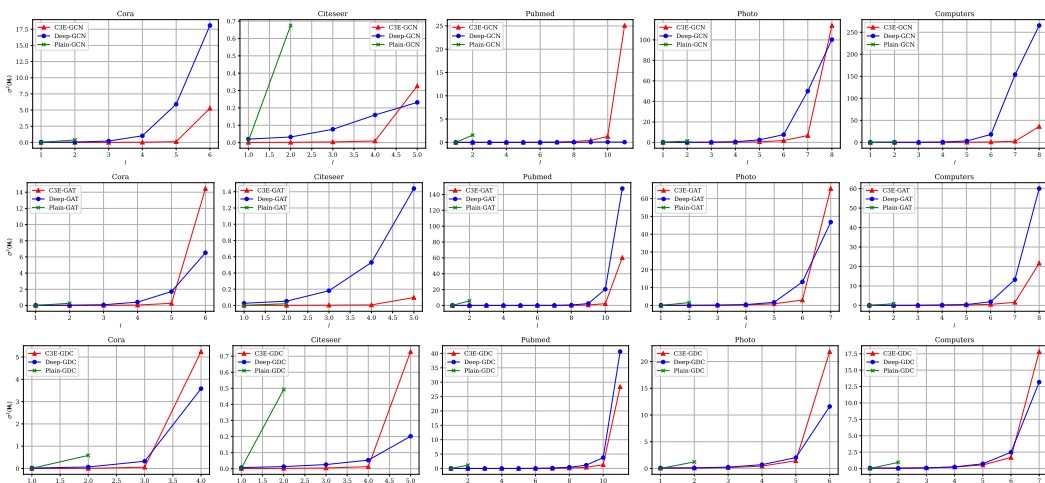

Figure 4: The layer-wise changes of $\sigma^2(\mathbf{H}_l)$ of C3E variants, the Deep variants, and plain backbone models. $\sigma^2(\mathbf{H}_l)$ increases substantially at the last layer (linear-reshaping layer) due to $w_L\sigma^2(\mathbf{H}_L)$. More detailed transitions can be found in Fig. 7 of Appendix. D.

in the learned node representation. Notably, the graph attention mechanism indeed helps models to avoid learning biased node representation, as $\mathbb{E}(\mathbf{H}_l)$ of all graph attention models are smaller than others. Regarding $\sigma^2(\mathbf{H}_l)$, from Fig. 4, we can observe that it increases gradually as the layer goes deeper. Moreover, the magnitudes and shapes of $\sigma^2(\mathbf{H}_l)$ curves can vary considerably to the input graph and information propagation methods. This implies that besides the widths and depth, graph properties such as $n$ and $\sigma^2(\tilde{\mathbf{A}})$ pose significant impacts on the model. In addition, we provide layer-wise mean and variance of node feature vectors $H_l^i$ in Appendix D to further verify our proposed theory.

## 8 DISCUSSION

GNNs learn meaningful representations from graph structures and attribute features by propagating various graph signals across the graphs. Nevertheless, many studies often neglect the significance of learnable matrices' dimensionalities and network depth. In this paper, we show that besides intricate information propagation mechanisms, the learning abilities of GNNs are highly dependent on their widths and depths.

Empirically, we demonstrate that GNNs perform similarly to noisy communication channels, reaching their optimal information transmission states when Shannon's theorem is satisfied and degenerating into band-limited channels when not. The required widths and depths of the network to achieve this critical phase transition are related to various properties of input graphs, even graphs with the same number of nodes or edges can vary considerably. Furthermore, reaching the optimal information transmission state does not necessarily guarantee improvements in model performance. This is illustrated in the Appendix F that C3E variants might still suffer from some inevitable problems of GNNs. Additionally, we provide theoretical analysis and explanations of favored operations in GNNs such as residual connection (Appendix B.1) and graph rewiring (Appendix B.2) based on our proposed theory.

Despite these promising results and theories, this work has several limitations. First, we consider the maximum entropy state of GNNs, such that the consideration of activation functions is bypassed. Nonetheless, activation functions have crucial impacts on the distribution of latent node representation, which can be observed in Fig. 3 and Fig. 4 that $\mathbb{E}(\mathbf{H}_l)$ is close to zero but not exactly zero and the magnitude of $\sigma^2(\mathbf{H}_l)$ is relatively smaller. Second, in an attempt to avoid biased results, other GNNs that do not take the form of Eq. (7) and other aggregation methods are not included. It is of interest to complete the formal theories, making it possible to generalize C3E to the broader family of GNNs.

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

## A APPENDIX

### A.1 PRECONDITIONS

Given a real-valued unknown distribution supported on $(-\infty, \infty)$, its maximum entropy is reached by its corresponding Gaussian distribution with mean $\mu$ and variance $\sigma^2$. Therefore, we can assume independence between variables under the principle of maximum entropy.

**Lemma 1.** *Given a real-valued random variable* $\mathbf{S}$, *its entropy* $H(\mathbf{S})$ *is bounded by the entropy of the corresponding Gaussian distribution* $\mathcal{N}(\mu, \sigma^2)$,

$$
\begin{aligned}
H(\mathbf{S}) &\leq -\int_{-\infty}^{\infty} \frac{1}{\sqrt{2\pi\sigma^2}} e^{-\frac{(s-\mu)^2}{2\sigma^2}} \log\left(\frac{1}{\sqrt{2\mu\sigma^2}} e^{-\frac{(s-\mu)^2}{2\sigma^2}}\right) ds \\
&\leq -\int_{-\infty}^{\infty} \frac{1}{\sigma\sqrt{2\pi}} e^{-\frac{(s-\mu)^2}{2\sigma^2}} \left(\log\left(\frac{1}{\sigma\sqrt{2\pi}}\right) + \log\left(e^{-\frac{(s-\mu)^2}{2\sigma^2}}\right)\right) ds \\
&\leq -\int_{-\infty}^{\infty} \frac{1}{\sigma\sqrt{2\pi}} e^{-\frac{(s-\mu)^2}{2\sigma^2}} \left(-\log(\sigma) - \frac{1}{2}\log(2\pi) - \frac{(s-\mu)^2}{2\sigma^2}\right) ds \\
&\leq \int_{-\infty}^{\infty} \frac{1}{\sigma\sqrt{2\pi}} e^{-\frac{(s-\mu)^2}{2\sigma^2}} \left(\log(\sigma) + \frac{1}{2}\log(2\pi) + \frac{(s-\mu)^2}{2\sigma^2}\right) ds \\
&\leq \log(\sigma) \int_{-\infty}^{\infty} \frac{1}{\sigma\sqrt{2\pi}} e^{-\frac{(s-\mu)^2}{2\sigma^2}} ds + \frac{1}{2}\log(2\pi) \int_{-\infty}^{\infty} \frac{1}{\sigma\sqrt{2\pi}} e^{-\frac{(s-\mu)^2}{2\sigma^2}} ds \\
&\quad + \frac{1}{2\sigma^2} \int_{-\infty}^{\infty} \frac{(s-\mu)^2}{\sigma\sqrt{2\pi}} e^{-\frac{(s-\mu)^2}{2\sigma^2}} ds \\
&\leq \log(\sigma) \cdot 1 + \frac{1}{2}\log(2\pi) \cdot 1 + \frac{1}{2\sigma^2} \cdot \sigma^2 \cdot 1 \\
&\leq \frac{1}{2}\log(2\pi e\sigma^2)
\end{aligned}
\tag{20}
$$

From a statistical physics perspective, the principle of maximum entropy suggests that the equilibrium state of a system should be taken in the state of distribution where the entropy of the system is at its maximum (Feynman, 2018). Assuming the equilibrium state of a physical system corresponds to the optimal state in an information system, the entropy of the optimal state can be expressed as the upper bound of the entropy value (Jaynes, 2003). Consequently, $H(\mathbf{S})$ is defined by its maxima under such circumstance,

$$
H(\mathbf{S}) \triangleq \frac{1}{2}\log(2\pi e\sigma^2)
\tag{21}
$$

**Lemma 2.** *For a set of random variables* $\{\mathbf{S}_1, \mathbf{S}_2, \ldots\}$, *their mathematical expectation and variance have following relationships,*

$$
\mathbb{E}(\sum_{i=1} \mathbf{S}_i) = \sum_{i=1} \mathbb{E}(\mathbf{S}_i), \ \sigma^2(\sum_{i=1} \mathbf{S}_i) = \sum_{i=1} \sigma^2(\mathbf{S}_i)
\tag{22}
$$

*The laws of expectation and variance of the product of the set of random variables are*

$$
\mathbb{E}(\prod_{i=1} \mathbf{S}_i) = \prod_{i=1} \mathbb{E}(\mathbf{S}_i),
\tag{23}
$$

$$
\sigma^2(\mathbf{S}_i\mathbf{S}_j) = \sigma^2(\mathbf{S}_i)\sigma^2(\mathbf{S}_j) + \sigma^2(\mathbf{S}_i)\mathbb{E}^2(\mathbf{S}_j) + \mathbb{E}^2(\mathbf{S}_i)\sigma^2(\mathbf{S}_j).
\tag{24}
$$

### A.2 ENTROPY OF GRAPH NEURAL NETWORKS

Directly measuring the entropy of the neural network is neither feasible nor desirable due to the massive interactions within the network, especially given that there are no widely established methods or theories for measuring the entropy of a neural network as a whole. Prior studies (Kingma, 2013; Rezende et al., 2014) point out that latent variables or learned representations should ideally follow a specific distribution $\Theta$, which implies that $\mathbf{H}_l \sim \Theta$. Fortunately, we can approximate the entropy of a neural network by its generated latent representation (Chan et al., 2022; Shen et al.,

2023), which is essentially a probability distribution reflecting the channel capacity of the network. In an ideally well-trained neural network, the output is the result of transforming the initial features through a series of nonlinear functions. If the latent representation has higher entropy, it suggests that the network has retained or introduced more information, and vice versa (Alemi et al., 2022). In other words, this indicates the uncertainty of the network or the spread of information encoded by the entire system. Hence, the entropy of a GNN is approximated by the entropy of its generated latent node representation in this work.

We have the following relationship based on the principle of maximum entropy as stated previously,

$$
\begin{aligned}
\boldsymbol{H}_l &= \boldsymbol{U}_l \boldsymbol{W}_l \\
&= (\tilde{\boldsymbol{A}} \boldsymbol{H}_{l-1}) \boldsymbol{W}_l \\
&= \tilde{\boldsymbol{A}} \left( \tilde{\boldsymbol{A}} \left( \cdots \left( \tilde{\boldsymbol{A}} \left( \tilde{\boldsymbol{A}} \boldsymbol{H}_0 \boldsymbol{W}_1 \right) \boldsymbol{W}_2 \right) \cdots \right) \boldsymbol{W}_{l-1} \right) \boldsymbol{W}_l.
\end{aligned}
\tag{25}
$$

Then, assume $\mathbf{W}_l \sim \mathcal{N}(0, 1)$ and $\mathbf{H}_0 \sim \mathcal{N}(0, 1)$, the mathematical expectation of $\mathbf{H}_l$ is defined by the following based on Eq. (22), Eq. (23), Eq.(24) and Eq. (25),

$$
\begin{aligned}
\mathbb{E}(\mathbf{H}_l) &= \mathbb{E}(\boldsymbol{H}_l^{i,j}) \\
&= \mathbb{E}(\sum_{o=1}^{w_{l-1}} \boldsymbol{W}_l^{o,j} \boldsymbol{U}_l^{i,o}) \\
&= \sum_{j=1}^{w_{l-1}} \mathbb{E}(\boldsymbol{W}_l^{o,j} \boldsymbol{U}_l^{i,o}) \\
&= \sum_{j=1}^{w_{l-1}} 0 \cdot \mathbb{E}(\boldsymbol{U}_l^{i,o}) \\
&= 0.
\end{aligned}
\tag{26}
$$

Similarly, the variance of $\mathbf{H}^{i,j}$ is defined by,

$$
\begin{aligned}
\sigma^2(\mathbf{H}_l) &= \sigma^2(\boldsymbol{H}_l^{i,j}) \\
&= \sigma^2(\sum_{o=1}^{w_{l-1}} \boldsymbol{W}_l^{o,j} \boldsymbol{U}_l^{i,o}) \\
&= \sum_{o=1}^{w_{l-1}} \sigma^2(\boldsymbol{W}_l^{o,j}) \sigma^2(\boldsymbol{U}_l^{i,o}) + \sigma^2(\boldsymbol{U}_l^{i,o}) \mathbb{E}^2(\boldsymbol{W}_l^{o,j}) + \mathbb{E}^2(\boldsymbol{U}_l^{i,o}) \sigma^2(\boldsymbol{W}_l^{o,j}) \\
&= \sum_{o=1}^{w_{l-1}} \sigma^2(\boldsymbol{W}_l^{o,j}) \sigma^2(\boldsymbol{U}_l^{i,o}) + \mathbb{E}^2(\boldsymbol{U}_l^{i,o}) \\
&= \sum_{o=1}^{w_{l-1}} \sigma^2(\boldsymbol{U}_l^{i,o}) \\
&= w_{l-1} \sigma^2(\boldsymbol{U}_l^{i,o}) \\
&= \prod_{o=0}^{l-1} n w_o \sigma^2(\tilde{\mathbf{A}}_o) \cdot 1^2.
\end{aligned}
\tag{27}
$$

Based on Eq. (20) and Eq. (21), the entropy of $\mathbf{H}_L$ is defined by,

$$
H(\mathbf{H}_L) = \frac{1}{2} \log(2\pi e) + \frac{1}{2} \sum_{l=0}^{L-1} \log(w_l) + \frac{1}{2} \sum_{l=0}^{L-1} \log(n\sigma^2(\tilde{\mathbf{A}}_l)).
\tag{28}
$$

Assume $\boldsymbol{D}^{ii} \approx d$, $n \gg d$, and $\tilde{\boldsymbol{A}}_l = \tilde{\boldsymbol{A}} = \boldsymbol{D}^{-\frac{1}{2}} \boldsymbol{A} \boldsymbol{D}^{-\frac{1}{2}}$. Then, $\sigma^2(\tilde{\mathbf{A}}_l)$ becomes,

$$
\sigma^2(\tilde{\mathbf{A}}_l) \approx \sigma^2(\frac{\mathbf{A}}{\sqrt{dd}}) = \frac{1}{d^2} \sigma^2(\mathbf{A}).
\tag{29}
$$

Then, substituting Eq. (29) into Eq. (28), $H(\mathbf{H}_L)$ becomes,

$$H(\mathbf{H}_L) \approx \frac{1}{2}\log{(2\pi e)} + \frac{1}{2}\sum_{l=0}^{L-1}\log{(w_l)} + \frac{1}{2}\sum_{l=0}^{L-1}\log{(\frac{n}{d^2}\sigma^2(\mathbf{A}))}. \tag{30}$$

Following previous studies (You et al., 2020), a linear layer as post-processing produces better results. Therefore, the entropy of an $L$-layer GNN ultimately becomes,

$$H(\Omega_L) = \frac{1}{2}\log{(2\pi e w_L \sigma^2(\mathbf{H}_l))}$$

$$= \frac{1}{2}\log{(2\pi e)} + \frac{1}{2}\log{(w_L)} + \frac{1}{2}\sum_{l=0}^{L-1}\log{(w_l)} + \frac{1}{2}\sum_{l=0}^{L-1}\log{(\frac{n}{d^2}\sigma^2(\mathbf{A}))}. \tag{31}$$

Nonetheless, real-world graphs exhibit various properties in terms of graph structures. For simplicity and unbiased estimation, Erdős–Rényi–Gilbert random graph (Erdos et al., 1960) can be adopted to approximate $\sigma^2(\mathbf{A})$ as a null model. Hence, $H(\Omega_L)$ can be further expressed as,

$$H(\Omega_L) \approx \frac{1}{2}\log{(2\pi e)} + \frac{1}{2}\log{(w_L)} + \frac{1}{2}\sum_{l=0}^{L-1}\log{(w_l)} + \frac{1}{2}\sum_{l=0}^{L-1}\log{(\frac{n}{d^2}\frac{d}{n}(1-\frac{d}{n}))}$$

$$\approx \frac{1}{2}\log{(2\pi e)} + \frac{1}{2}\log{(w_L)} + \sum_{l=0}^{L-1}\frac{1}{2}\log{(\frac{w_l}{d})}. \tag{32}$$

## A.3 Channel Capacity of Graph Neural Networks

Theoretically, an ideal GNN learns the exact latent mapping between $\mathcal{G}'$ and $\mathcal{G}$. Consequently, we have $H(\mathcal{G}'|\Omega_L(\mathcal{G})) = H(\Omega_L(\mathcal{G})|\mathcal{G}') = 0$. Hence, $\phi(\Omega_L)$ can be rewritten as,

$$\phi(\Omega_L) = H(\Omega_L(\mathcal{G})) - H(\Omega_L(\mathcal{G})|\mathcal{G}') = H(\mathbf{H}_L). \tag{33}$$

Furthermore, the following inequation holds if $H(\mathbf{H}_0) \geq 0$,

$$\phi(\Omega_L) \geq \phi(\Omega_L) - H(\mathbf{H}_0)$$

$$\geq \sum_{l=1}^{L} H(\mathbf{H}_l) - H(\mathbf{H}_{l-1})$$

$$\geq \sum_{l=1}^{L} \frac{H(\mathbf{H}_l) - H(\mathbf{H}_{l-1})}{H(\mathbf{H}_{l-1})} H(\mathbf{H}_{l-1})$$

Nonetheless, this is the behavior of ideal neural networks. Considering real-world neural networks can never reach this critical point (i.e., $H(\mathbf{H}_{l-1}) \geq I(\mathbf{H}_l; \mathbf{H}_{l-1}) \geq 0$),

$$\phi(\Omega_L) \geq \sum_{l=1}^{L} \frac{H(\mathbf{H}_l) - H(\mathbf{H}_{l-1})}{H(\mathbf{H}_{l-1})} I(\mathbf{H}_l; \mathbf{H}_{l-1})$$

$$\geq \sum_{l=1}^{L} \frac{H(\mathbf{H}_l) - H(\mathbf{H}_{l-1})}{H(\mathbf{H}_{l-1})} (H(\mathbf{H}_{l-1}) + H(\mathbf{H}_l) - H(\mathbf{H}_l, \mathbf{H}_{l-1}))$$

Assume $w_o = 2 + \epsilon \geq 2$ and $w_k = 2 + \delta \geq 2$, we have the following relationship,

$$\epsilon + \delta + \delta\epsilon \geq 0,$$
$$4 + 2\epsilon + 2\delta + \delta\epsilon \geq 4 + \epsilon + \delta$$
$$(2 + \epsilon)(2 + \delta) \geq 2 + \epsilon + 2 + \delta$$
$$w_o w_k \geq w_o + w_k$$
$$w_l w_{l-1} \geq w_l + w_{l-1}$$
$$\log{(w_l)} + \log{(w_{l-1})} \geq \log{(w_l + w_{l-1})} \tag{34}$$

Previous works (He et al., 2016; Huang et al., 2017; Li et al., 2021) suggest that the neural network refines and compresses the information as depth increases with decreasing width,

$$w_1 \geq w_2 \cdots \geq w_{L-1} \geq w_L. \tag{35}$$

According to Eq. (28), Eq. (30), Eq. (34), and Eq. (35) the inequation becomes,

$$
\begin{aligned}
\phi(\Omega_L) &\geq \sum_{l=1}^{L} \frac{H(\mathbf{H}_l) - H(\mathbf{H}_{l-1})}{H(\mathbf{H}_{l-1})} \left( \log(w_l) + \log(w_{l-1}) - \log(w_l + w_{l-1}) \right) \\
&\geq \sum_{l=1}^{L} \frac{H(\mathbf{H}_l) - H(\mathbf{H}_{l-1})}{H(\mathbf{H}_{l-1})} \log\left( \frac{w_{l-1} w_l}{w_{l-1} + w_l} \right) \\
&\geq \sum_{l=1}^{L} \frac{\log\left( \prod_{o=0}^{l-1} \frac{n w_o}{d^2} \sigma^2(\mathbf{A}) \right) - \log\left( \prod_{o=0}^{l-2} \frac{n w_o}{d^2} \sigma^2(\mathbf{A}) \right)}{\log\left( \prod_{o=0}^{l-2} \frac{n w_o}{d^2} \sigma^2(\mathbf{A}) \right) + \log(2\pi e)} \log\left( \frac{w_{l-1} w_l}{w_{l-1} + w_l} \right) \\
&\geq \sum_{l=1}^{L} \frac{\log\left( \frac{n w_{l-1}}{d^2} \sigma^2(\mathbf{A}) \right)}{\log(2\pi e) + \sum_{o=0}^{l-2} \log\left( \frac{n w_o}{d^2} \sigma^2(\mathbf{A}) \right)} \log\left( \frac{w_{l-1} w_l}{w_{l-1} + w_l} \right) \\
&\geq \sum_{l=1}^{L} \frac{\log\left( \frac{w_{l-1}}{d} \right)}{\log(2\pi e) + \sum_{o=0}^{l-2} \log\left( \frac{w_o}{d} \right)} \log\left( \frac{w_{l-1} w_l}{w_{l-1} + w_l} \right) \\
&\geq \sum_{l=1}^{L} \frac{1}{\log_{\frac{w_{l-1}}{d}}(2\pi e) + \sum_{o=0}^{l-2} \log_{\frac{w_{l-1}}{d}}\left( \frac{w_o}{d} \right)} \log\left( \frac{w_{l-1} w_l}{w_{l-1} + w_l} \right) \\
&\geq \sum_{l=1}^{L} \frac{1}{\log_{\frac{w_{l-1}}{d}}(2\pi e) + \sum_{o=0}^{l-2} 1} \log\left( \frac{w_{l-1} w_l}{w_{l-1} + w_l} \right)
\end{aligned}
\tag{36}
$$

In practice, the average degrees of many real-world graphs are not too large. For example, $d_{\text{Cora}} \approx 3.9$, $d_{\text{Citeseer}} \approx 2.7$, and $d_{\text{Pubmed}} \approx 4.5$. The feasible solutions of $w_l$ typically range from $2 \times 10^2$ to $5 \times 10^3$ and $2\pi e \approx 17.0795$, such that $\log_{\frac{w_{l-1}}{d}}(2\pi e) \leq 1$. Then, we have the following relationship,

$$\phi(\Omega_L) \geq \sum_{l=1}^{L} \frac{1}{l} \log\left( \frac{w_{l-1} w_l}{w_{l-1} + w_l} \right). \tag{37}$$

# B PRACTICAL OPERATIONS IN GRAPH NEURAL NETWORKS

## B.1 RESIDUAL CONNECTION IN GRAPH NEURAL NETWORK

Residual connection is another favored technique used in deep learning, which is generally defined as,

$$\boldsymbol{H}_l = f(\boldsymbol{H}_{l-1}) + \boldsymbol{H}_{l-1} \tag{38}$$

Here, $f(\cdot)$ denotes the corresponding layer update function. Consequently, $\mathbb{E}(\mathbf{H}_l)$ becomes,

$$
\begin{aligned}
\mathbb{E}(\mathbf{H}_l) &= \mathbb{E}(\boldsymbol{H}_{l-1}^{i,j} + \sum_{o=1}^{w_{l-1}} \boldsymbol{W}_l^{o,j} \boldsymbol{U}_l^{i,o}) \\
&= \mathbb{E}(\mathbf{H}_{l-1}) + \sum_{j=1}^{w_{l-1}} \mathbb{E}(\boldsymbol{W}_l^{o,j} \boldsymbol{U}_l^{i,o}) \\
&= 0 + \sum_{j=1}^{w_{l-1}} 0 \cdot \mathbb{E}(\boldsymbol{U}_l^{i,o}) \\
&= 0.
\end{aligned}
\tag{39}
$$

Meanwhile, $\sigma^2(\mathbf{H}_l)$ becomes,

$$
\begin{aligned}
\sigma^2(\mathbf{H}_l) &= \sigma^2(\boldsymbol{H}_{l-1}^{i,j} + \sum_{o=1}^{w_{l-1}} \boldsymbol{W}_l^{o,j} \boldsymbol{U}_l^{i,o}) \\
&= \sigma^2(\mathbf{H}_{l-1}) + \sigma^2(\sum_{o=1}^{w_{l-1}} \boldsymbol{W}_l^{o,j} \boldsymbol{U}_l^{i,o}) + 2\mathrm{Cov}(\boldsymbol{H}_{l-1}^{i,j}, \sum_{o=1}^{w_{l-1}} \boldsymbol{W}_l^{o,j} \boldsymbol{U}_l^{i,o}) \\
&= \prod_{o=0}^{l-2} nw_o\sigma^2(\tilde{\mathbf{A}}_o) + \prod_{o=0}^{l-1} nw_o\sigma^2(\tilde{\mathbf{A}}_o) + 2\mathrm{Cov}(\boldsymbol{H}_{l-1}^{i,j}, \sum_{o=1}^{w_{l-1}} \boldsymbol{W}_l^{o,j} \boldsymbol{U}_l^{i,o}) \\
&= \prod_{o=0}^{l-2} nw_o\sigma^2(\tilde{\mathbf{A}}_o) + \prod_{o=0}^{l-1} nw_o\sigma^2(\tilde{\mathbf{A}}_o) - 2\mathbb{E}(\boldsymbol{H}_{l-1}^{i,j} \cdot \sum_{o=1}^{w_{l-1}} \boldsymbol{W}_l^{o,j} \boldsymbol{U}_l^{i,o}) \\
&= \prod_{o=0}^{l-2} nw_o\sigma^2(\tilde{\mathbf{A}}_o) + \prod_{o=0}^{l-1} nw_o\sigma^2(\tilde{\mathbf{A}}_o) - 2\mathbb{E}(\sum_{o=1}^{w_{l-1}} \boldsymbol{W}_l^{o,j} \boldsymbol{U}_l^{i,o} \boldsymbol{H}_{l-1}^{i,j}) \\
&= \prod_{o=0}^{l-2} nw_o\sigma^2(\tilde{\mathbf{A}}_o) + \prod_{o=0}^{l-1} nw_o\sigma^2(\tilde{\mathbf{A}}_o) - 2\sum_{o=1}^{w_{l-1}} \mathbb{E}(\boldsymbol{W}_l^{o,j} \boldsymbol{U}_l^{i,o} \boldsymbol{H}_{l-1}^{i,j}) \\
&= \prod_{o=0}^{l-2} nw_o\sigma^2(\tilde{\mathbf{A}}_o) + \prod_{o=0}^{l-1} nw_o\sigma^2(\tilde{\mathbf{A}}_o) - 2\sum_{o=1}^{w_{l-1}} 0 \cdot \mathbb{E}(\boldsymbol{U}_l^{i,o} \boldsymbol{H}_{l-1}^{i,j}) \\
&= \prod_{o=0}^{l-2} nw_o\sigma^2(\tilde{\mathbf{A}}_o) + \prod_{o=0}^{l-1} nw_o\sigma^2(\tilde{\mathbf{A}}_o).
\end{aligned}
\tag{40}
$$

According to Eq. (32), Eq. (37), Eq. (39), and Eq. (40), adopting residual connection results in increasing model entropy and model channel capacity. Thus, residual connection offers great potential in augmenting the model performance of relatively deep GNNs. If Eq. (38) becomes the following form,

$$
\boldsymbol{H}_l = f(\boldsymbol{H}_{l-1}) + \mathcal{T}(\boldsymbol{H}_{l-1}),
\tag{41}
$$

where $\mathcal{T}(\cdot)$ is a linear transform without nonlinear activation, then Eq. (39) and Eq. (40) still hold. According to Eq. (39) and Eq. (40), $H(\Omega_L)$ becomes,

$$
\begin{aligned}
H(\Omega_L) &= \frac{1}{2} \log\left(2\pi e w_L \sigma^2(\mathbf{H}_l)\right) \\
&= \frac{1}{2} \log\left(2\pi e\right) + \frac{1}{2} \log\left(w_L\right) + \frac{1}{2} \sum_{l=0}^{L-1} \log\left(w_l\right) + \frac{1}{2} \sum_{l=0}^{L-1} \log\left(\frac{n}{d^2}\sigma^2(\mathbf{A})\right) \\
&\quad + \frac{1}{2} \sum_{l=0}^{L-2} \log\left(w_l\right) + \frac{1}{2} \sum_{l=0}^{L-2} \log\left(\frac{n}{d^2}\sigma^2(\mathbf{A})\right) \\
&\approx \frac{1}{2} \log\left(2\pi e\right) + \frac{1}{2} \log\left(w_L\right) + \sum_{l=0}^{L-1} \frac{1}{2} \log\left(\frac{w_l}{d}\right) + \sum_{l=0}^{L-2} \frac{1}{2} \log\left(\frac{w_l}{d}\right).
\end{aligned}
\tag{42}
$$

Then, the lower bound of $\phi(\Omega_L)$ becomes,

$$
\begin{aligned}
\phi(\Omega_L) &\geq \sum_{l=1}^{L} \frac{H(\mathbf{H}_l) - H(\mathbf{H}_{l-1})}{H(\mathbf{H}_{l-1})} \log\left(\frac{w_{l-1}w_l}{w_{l-1} + w_l}\right) \\
&\geq \sum_{l=1}^{L} \frac{\log\left(\prod_{o=0}^{l-1} \frac{w_o}{d}\right) - \log\left(\prod_{o=0}^{l-3} \frac{w_o}{d}\right)}{\log\left(\prod_{o=0}^{l-2} \frac{w_o}{d}\right) + \log\left(\prod_{o=0}^{l-3} \frac{w_o}{d}\right) + \log\left(2\pi e\right)} \log\left(\frac{w_{l-1}w_l}{w_{l-1} + w_l}\right) \\
&\geq \sum_{l=1}^{L} \frac{\log\frac{w_{l-1}}{d} + \log\frac{w_{l-2}}{d}}{\log\left(2\pi e\right) + \log\left(\prod_{o=0}^{l-2} \frac{w_o}{d}\right) + \log\left(\prod_{o=0}^{l-3} \frac{w_o}{d}\right)} \log\left(\frac{w_{l-1}w_l}{w_{l-1} + w_l}\right)
\end{aligned}
$$

$$\geq \sum_{l=1}^{L} \frac{1}{\log \frac{w_{l-1} w_{l-2}}{d^2} \left(2\pi e \frac{w_0}{d}\right) + \sum_{o=1}^{l-2} \log \frac{w_{l-1} w_{l-2}}{d^2} \left(\frac{w_o w_{o-1}}{d^2}\right)} \log \left(\frac{w_{l-1} w_l}{w_{l-1} + w_l}\right)$$

$$\geq \sum_{l=1}^{L} \frac{1}{\log \frac{w_{l-1} w_{l-2}}{d^2} (2\pi e) + \log \frac{w_{l-1} w_{l-2}}{d^2} \left(\frac{w_0}{d}\right) + \sum_{o=1}^{l-2} 1} \log \left(\frac{w_{l-1} w_l}{w_{l-1} + w_l}\right)$$

$$\geq \sum_{l=1}^{L} \frac{1}{l} \log \left(\frac{w_{l-1} w_l}{w_{l-1} + w_l}\right). \tag{43}$$

Accordingly, we can observe that $H(\Omega_L)$ increases as $H(\mathbf{H}_l)$ increases and the first two terms in the denominator are smaller than in Eq. (36), which leads to an increase of $\phi(\Omega_L)$. Besides this, the lower bound of $\phi(\Omega_L)$ remains unchanged, ensuring Shannon's theorem is satisfied.

## B.2 ADJACENCY INFORMATION IN GRAPH NEURAL NETWORK

Nowadays, graph rewiring has been a common and popular technique to enhance the performance of GNNs, which commonly includes dropping edge (Rong et al., 2020), graph rewiring (Barbero et al., 2024), etc. According to Eq. (28), Eq. (32), and Eq. (37), altering the adjacency matrix can pose significant impacts on the model entropy and the model channel capacity. Given a defined GNN model and the fixed number of nodes, we can obtain the partial derivate $\frac{\partial H(\Omega_L)}{\partial \sigma^2(\mathbf{A})}$ based on Eq. (31),

$$\frac{\partial H(\Omega_L)}{\partial \sigma^2(\mathbf{A})} = \sum_{l=0}^{L-1} \frac{\partial}{\partial \sigma^2(\mathbf{A})} \frac{1}{2} \log \left(\frac{n}{d^2} \sigma^2(\mathbf{A})\right)$$

$$= \frac{1}{2} \sum_{l=0}^{L-1} \frac{1}{\frac{n}{d^2} \sigma^2(\mathbf{A})} \cdot \frac{n}{d^2}$$

$$= \frac{1}{2} \sum_{l=0}^{L-1} \frac{1}{\sigma^2(\mathbf{A})}$$

$$= \frac{L}{2\sigma^2(\mathbf{A})}. \tag{44}$$

The partial derivate $\frac{\partial H(\Omega_L)}{\partial d}$ is obtained by,

$$\frac{\partial H(\Omega_L)}{\partial d} = \frac{1}{2} \sum_{l=0}^{L-1} \frac{\partial}{\partial d} \log \left(\frac{n}{d^2} \sigma^2(\mathbf{A})\right)$$

$$= \frac{1}{2} \sum_{l=0}^{L-1} \left(-\frac{2}{d} \cdot \frac{1}{\frac{n}{d^2} \sigma^2(\mathbf{A})}\right)$$

$$= -\frac{1}{d} \sum_{l=0}^{L-1} \frac{1}{\sigma^2(\mathbf{A})}$$

$$= -\frac{L}{d}. \tag{45}$$

Given a defined GNN model and the fixed number of nodes, the partial derivative $\frac{\partial \phi_0(\Omega_L)}{\partial \sigma^2(\mathbf{A})}$ is obtained by,

$$\frac{\partial \phi_0(\Omega_L)}{\partial \sigma^2(\mathbf{A})} = \sum_{l=1}^{L} \frac{\partial}{\partial \sigma^2(\mathbf{A})} \left[\frac{1}{\log \frac{n w_{l-1}}{d^2} \sigma^2(\mathbf{A})} (2\pi e) + \sum_{o=0}^{l-2} 1} \log \left(\frac{w_{l-1} w_l}{w_{l-1} + w_l}\right)\right]$$

$$= \sum_{l=1}^{L} \left[ \frac{\log\left(2\pi e\right)\log\left(\frac{w_{l-1}w_l}{w_l+w_{l-1}}\right)\right) d^2 \cdot \frac{nw_{l-1}}{d^2} \cdot \frac{\partial}{\partial\sigma^2(\mathbf{A})}\sigma^2(\mathbf{A})}{nw_{l-1}\sigma^2(\mathbf{A})\left(\frac{\log(2\pi e)}{\log\left(\frac{nw_{l-1}\sigma^2(\mathbf{A})}{d^2}\right)}+l-1\right)^2 \log^2\left(\frac{nw_{l-1}\sigma^2(\mathbf{A})}{d^2}\right)} \right]$$

$$= \sum_{l=1}^{L} \frac{\log\left(2\pi e\right)\log\left(\frac{w_{l-1}w_l}{w_l+w_{l-1}}\right)}{\sigma^2(\mathbf{A})\left(\frac{\log(2\pi e)}{\log\left(\frac{nw_{l-1}\sigma^2(\mathbf{A})}{d^2}\right)}+l-1\right)^2 \log^2\left(\frac{nw_{l-1}\sigma^2(\mathbf{A})}{d^2}\right)}$$

$$= \sum_{l=1}^{L} \frac{\log\left(2\pi e\right)\log\left(\frac{w_{l-1}w_l}{w_l+w_{l-1}}\right)}{\sigma^2(\mathbf{A})\left((l-1)\log\left(\frac{nw_{l-1}\sigma^2(\mathbf{A})}{d^2}\right)+\log\left(2\pi e\right)\right)^2} \tag{46}$$

Similarly, the partial derivative $\frac{\partial\phi_0(\Omega_L)}{\partial d}$ is obtained by,

$$\frac{\partial\phi_0(\Omega_L)}{\partial d} = \sum_{l=1}^{L} \frac{\partial}{\partial d}\left[ \frac{1}{\log_{\frac{w_{l-1}}{d}}(2\pi e)+\sum_{o=0}^{l-2}1}\log\left(\frac{w_{l-1}w_l}{w_{l-1}+w_l}\right)\right]$$

$$= \sum_{l=1}^{L}\left[ -\frac{1}{\left(\log_{\frac{w_{l-1}}{d}}(2\pi e)+\sum_{o=0}^{l-2}1\right)^2} \cdot \frac{-\log(2\pi e)}{d\log^2\left(\frac{w_{l-1}}{d}\right)} \cdot \log\left(\frac{w_{l-1}w_l}{w_{l-1}+w_l}\right)\right]$$

$$= -\sum_{l=1}^{L} \frac{\log\left(\frac{w_{l-1}w_l}{w_{l-1}+w_l}\right)\cdot\frac{\log(2\pi e)}{d\log^2\left(\frac{w_{l-1}}{d}\right)}}{\left(\log_{\frac{w_{l-1}}{d}}(2\pi e)+\sum_{o=0}^{l-2}1\right)^2}$$

$$= -\sum_{l=1}^{L} \frac{\log\left(2\pi e\right)\cdot\log\left(\frac{w_{l-1}w_l}{w_{l-1}+w_l}\right)}{d\left((l-1)\log\left(\frac{w_{l-1}}{d}\right)+\log\left(2\pi e\right)\right)^2}. \tag{47}$$

We can observe that the model performance generally improves as the model entropy and the model channel capacity grow based on previous results. Consequently, approaches like graph rewiring (Jin et al., 2020; Barbero et al., 2024) and utilizing attention mechanisms can substantially enhance the performance of GNN models. Thus, performing graph rewiring and adopting attention-based mechanisms provides the opportunity to reduce $d$ or increase $\sigma^2(\mathbf{A})$ during the information propagation process.

## C DATASETS INFORMATION

Table 2: Citation networks including Cora, Citeseer, and Pubmed are sliced with public splits. Co-purchase graphs consisting of AmazonPhoto and AmazonComputers are sliced with random splits.

| Dataset | #Node | #Edges | # Features | #Classes | #Avg. Degree | Label Rate |
|---|---|---|---|---|---|---|
| Cora | 2708 | 5429 | 1433 | 7 | 3.9 | 0.0517 |
| Citeseer | 3312 | 4732 | 3703 | 6 | 7.0 | 0.0362 |
| Pubmed | 19717 | 44338 | 500 | 3 | 4.5 | 0.0030 |
| AmazonPhoto | 7650 | 119081 | 745 | 8 | 31.1 | 0.0209 |
| AmazonComputers | 13752 | 245861 | 767 | 10 | 35.8 | 0.0145 |

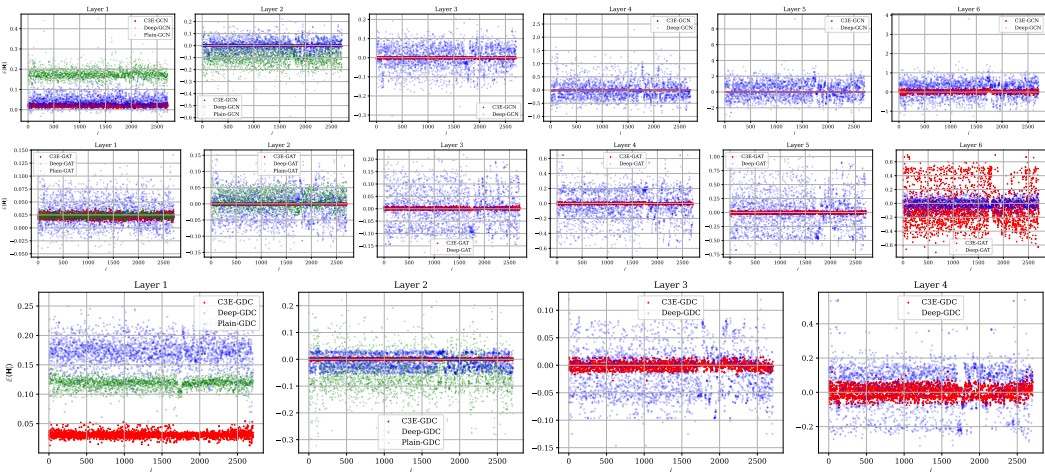

Figure 5: Mean of latent node feature vector by layer on Cora.

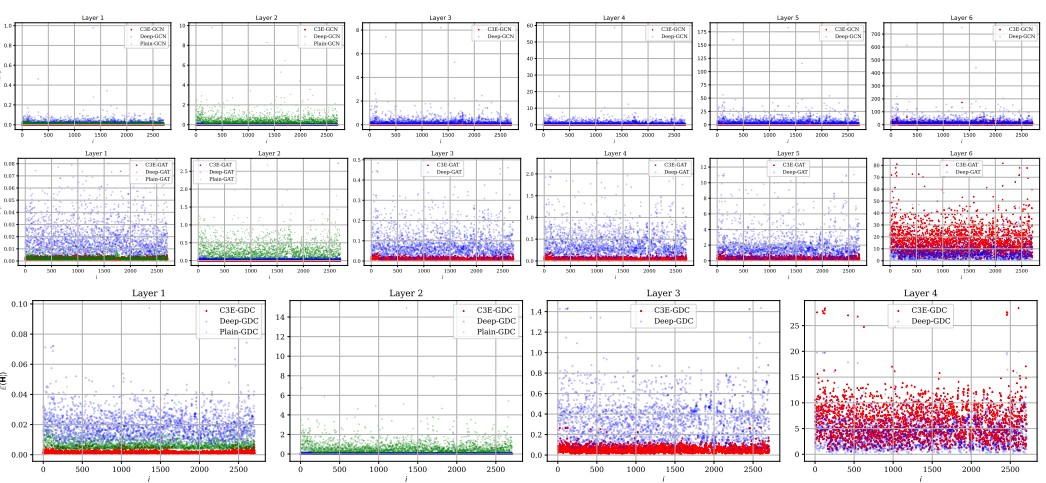

Figure 6: Variance of latent node feature vector by layer on Cora.

# D MEAN & VARIANCE OF NODES

Here, we present the mean and variance of latent node feature vectors in each layer of C3E variants, deep backbone models, and backbone models on Cora. From Fig. 5 and Fig. 6, we can observe that latent node feature vectors of C3E variants have the most stable and similar statistical properties in the message-passing layers as opposed to the deep backbone models, and backbone models. The mean and variance of latent node feature vectors of deep backbone models are even more versatile across layers, they can be centralized within a certain interval or dispersed to various intervals. These observations empirically validate our proposed theories and the finding of previous studies that latent variables or learned representations should ideally follow a specific distribution $\Theta$, i.e., $\mathbf{H}_l \sim \Theta$. Besides, we provide more detailed transition plots of variance of latent node representation in Fig. 7.

# E HYPERPARAMETER CONFIGURATION

We list the model configurations used in this work, no extra tricks or operations are implemented in this work.

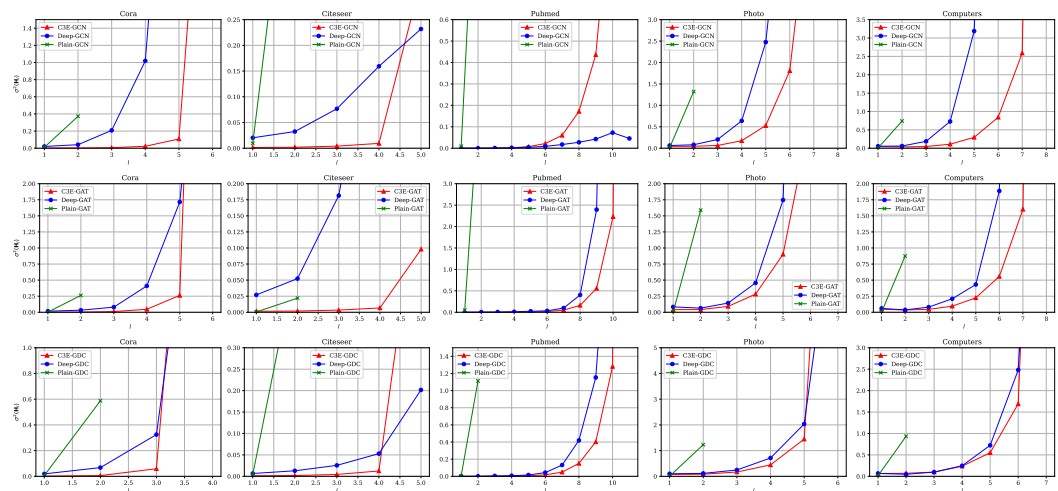

Figure 7: Detailed variance of latent node representation of Fig. 4.

### E.1 OBJECTIVE FUNCTION

The cross-entropy is selected as the objective function for performing the semi-supervised node classification task.

### E.2 OPTIMIZER SETTINGS

In this work, we use Adam/AdamW as the optimizer. The weight decay factors of backbone models and deep backbone models are set to the same values as the original works. The weight decay factors of C3E variants are set to $5 \times 10^{-4}$. The learning rates of backbone models and deep backbone models are set to the same values as the original works. The learning rates of C3E variants are searched from the interval $[0.9 \times 10^{-4}, 10^{-3}]$.

### E.3 RESIDUAL CONNECTION

For a fair comparison, only deep backbone models and C3E variants have residual connections in the form of Eq. (41). According to Eq. (43), to avoid being divided by zero, i.e., the layer index should satisfy $l \geq 2$, the residual connections are applied in all message-passing layers except for the first message-passing layers.

### E.4 DEPTH & WIDTHS

- Cora:
  - Plain-GCN($\{1433, 16, 7\}$),
  - Plain-GAT($\{1433, 64, 7\}$), head=8,
  - Plain-GDC($\{1433, 64, 7\}$),
    kernel=PPR, $\alpha = 0.05$, $k = 128$,
  - Deep-GCN($\{1433, 16, 16, 16, 16, 16, 7\}$),
  - Deep-GAT($\{1433, 64, 64, 64, 64, 64, 7\}$), head=8,
  - Deep-GDC($\{1433, 64, 64, 64, 7\}$),
    kernel=PPR, $\alpha = 0.05$, $k = 128$,
  - C3E-GCN($\{1433, 1982, 1298, 1094, 986, 722, 7\}$),
  - C3E-GAT($\{1433, 1982, 1290, 1064, 933, 720, 7\}$), head=2,
  - C3E-GDC($\{1433, 1488, 972, 688, 7\}$),
    kernel=PPR, $\alpha = 0.05$, $k = 128$.
- Citeseer:

- – Plain-GCN($\{3703, 16, 6\}$),
- – Plain-GAT($\{3703, 64, 6\}$), head=8,
- – Plain-GDC($\{3703, 64, 6\}$),
  kernel=PPR, $\alpha = 0.1$, $\epsilon = 9 \times 10^{-4}$,
- – Deep-GCN($\{3703, 16, 16, 16, 16, 6\}$),
- – Deep-GAT($\{3703, 64, 64, 64, 64, 6\}$), head=8,
- – Deep-GDC($\{3703, 64, 64, 64, 64, 6\}$),
  kernel=PPR, $\alpha = 0.1$, $\epsilon = 9 \times 10^{-4}$,
- – C3E-GCN($\{3703, 5270, 3430, 2861, 2124, 6\}$),
- – C3E-GAT($\{3703, 5270, 3432, 2862, 2126, 6\}$), head=2,
- – C3E-GDC($\{3703, 3988, 2598, 2164, 1610, 6\}$),
  kernel=PPR, $\alpha = 0.1$, $\epsilon = 9 \times 10^{-4}$.

- **Pubmed:**
  - – Plain-GCN($\{500, 16, 3\}$),
  - – Plain-GAT($\{500, 64, 3\}$), head=8,
  - – Plain-GDC($\{500, 64, 3\}$),
    kernel=PPR, $\alpha = 0.1$, $k = 64$,
  - – Deep-GCN($\{500, 16, 16, 16, 16, 16, 16, 16, 16, 16, 3\}$),
  - – Deep-GAT($\{500, 64, 64, 64, 64, 64, 64, 64, 64, 64, 3\}$), head=8,
  - – Deep-GDC($\{500, 64, 64, 64, 64, 64, 64, 64, 64, 64, 3\}$),
    kernel=PPR, $\alpha = 0.1$, $k = 64$,
  - – C3E-GCN($\{500, 682, 450, 370, 328, 300, 282, 266, 254, 248, 180, 3\}$),
  - – C3E-GAT($\{500, 682, 450, 370, 328, 300, 282, 266, 254, 248, 180, 3\}$), head=2,
  - – C3E-GDC($\{500, 534, 356, 298, 266, 244, 228, 216, 204, 194, 160, 3\}$),
    kernel=PPR, $\alpha = 0.1$, $k = 64$.

- **AmazonPhoto:**
  - – Plain-GCN($\{745, 16, 8\}$),
  - – Plain-GAT($\{745, 64, 8\}$), head=8,
  - – Plain-GDC($\{745, 64, 8\}$),
    kernel=PPR, $\alpha = 0.15$, $k = 64$
  - – Deep-GCN($\{745, 16, 16, 16, 16, 16, 16, 16, 8\}$),
  - – Deep-GAT($\{745, 64, 64, 64, 64, 64, 64, 8\}$), head=8,
  - – Deep-GDC($\{745, 64, 64, 64, 64, 64, 8\}$),
    kernel=PPR, $\alpha = 0.15$, $k = 64$
  - – C3E-GCN($\{745, 830, 538, 446, 400, 370, 350, 274, 8\}$),
  - – C3E-GAT($\{745, 830, 546, 458, 412, 384, 298, 8\}$), head=2,
  - – C3E-GDC($\{745, 796, 532, 444, 392, 308, 8\}$),
    kernel=PPR, $\alpha = 0.15$, $k = 64$.

- **AmazonComputers:**
  - – Plain-GCN($\{767, 16, 10\}$),
  - – Plain-GAT($\{767, 64, 10\}$), head=8,
  - – Plain-GDC($\{767, 64, 10\}$),
    kernel=PPR, $\alpha = 0.1$, $k = 64$,
  - – Deep-GCN($\{767, 16, 16, 16, 16, 16, 16, 16, 10\}$),
  - – Deep-GAT($\{767, 64, 64, 64, 64, 64, 64, 64, 10\}$), head=8,
  - – Deep-GDC($\{767, 64, 64, 64, 64, 64, 64, 10\}$),
    kernel=PPR, $\alpha = 0.1$, $k = 64$,
  - – C3E-GCN($\{767, 846, 554, 464, 416, 386, 366, 292, 10\}$),
  - – C3E-GAT($\{767, 846, 554, 464, 416, 386, 366, 292, 10\}$), head=2,
  - – C3E-GDC($\{767, 818, 546, 456, 407, 369, 293, 10\}$),
    kernel=PPR, $\alpha = 0.1$, $k = 64$.

### E.5 OTHER CONFIGURATIONS

According to Eq. 16, the dropout probability of C3E models is calculated by,

$$p_l = 1 - \frac{w_{l-1}}{w_{l-1} + w_l}. \tag{48}$$

The dropout probability of other models is 0.5 or 0.6 as their original works. Self-loop is enabled for all datasets. Empirically, we select PReLU$(\cdot)$ as the activation function (You et al., 2020) and only apply it after the first message-passing layer. Furthermore, we find that adding the nonlinear activation function to every layer hinders GNNs from effectively learning meaningful representation and degrades their performance. Based on experimental results, we propose a regularization term $\frac{w_l}{w_{l-1}} \le d^{\frac{1}{d}}$ to Eq. (18), and $\zeta = 1.3$.

## F DIRICHLET ENERGY ANALYSIS

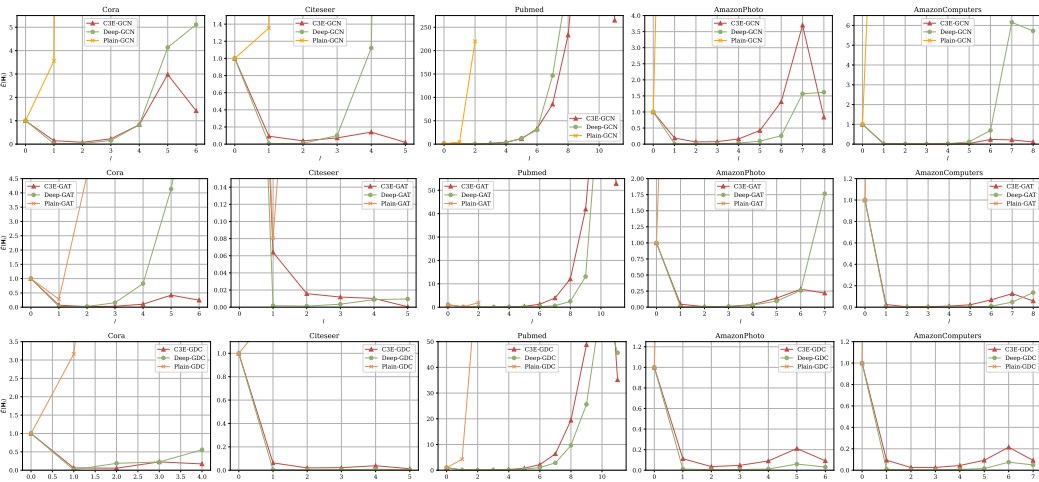

Figure 8: The transition of layer-wise normalized Dirichlet energy of node representation.

The Dirichlet energy is widely adopted in graph theory and graph signal processing to measure the smoothness of features. In this work, we use normalized Dirichlet Energy to quantify the changes in node representation regarding the corresponding initial node features. The normalized Dirichlet Energy is defined by,

$$E(\boldsymbol{H}_l) = \frac{\mathrm{tr}(\boldsymbol{H}_l^{\mathrm{T}} \mathcal{L} \boldsymbol{H}_l)}{\mathrm{tr}(\boldsymbol{X}^{\mathrm{T}} \mathcal{L} \boldsymbol{X})}, \tag{49}$$

where tr$(\cdot)$ denotes the trace and $\mathcal{L} = \boldsymbol{D} - \boldsymbol{A}$ denotes the graph laplacian matrix. From Fig. 8, we can observe that compared to the backbone models, the Dirichlet energy of Deep models and C3E variants reduces in the subsequent few layers, which might indicate the presence of over-smoothing. The Dirichlet energy of C3E models gradually decreases to a minimum value, then continuously increases to a certain value, and converges to a smaller value eventually. Although the Deep models sometimes show similar characteristics, they often diverge at deeper layers.

According to Zhou et al. (2021), the layer-wise Dirichlet energy should be constrained within certain lower bound and upper bound to avoid over-smoothing and over-separating. It points out that GNNs can stack more layers by constraining the Dirichlet energy of latent node representations, such that the Dirichlet energy rapidly decreases in the shallow layers, then consistently increases in the subsequent layers, and converges to a stable status (i.e., the Dirichlet energy almost remains the same) in deeper layers.

Based on these observations and previous results, we can conclude that having a small Dirichlet energy does not always mean poor model performance based on Fig. 8 and Table .1. In other words, if the benefit of reaching or approaching the optimal information transmission state is smaller than the inescapable negative effects (like over-smoothing or over-separating) of increasing the width and depth, then model performance is not guaranteed, and vice versa.

