# OpenReview forum: "Unleashing the Information Flow: Graph Neural Networks are Noisy Communication Channels"
_ICLR.cc/2025/Conference — ICLR 2025 Conference Withdrawn Submission_

### Official Review · Reviewer_e4SK · 2024-11-03

**Soundness:** 2
**Presentation:** 2
**Contribution:** 2
**Rating:** 3
**Confidence:** 4

**Summary:**

This paper concerns with the problem of choosing the widths and depth of message-passing GNNs for semi-supervised learning. A summary is as follows:
- In Section 4, assuming implicitly that maximum entropy state corresponds to the optimal information transmission rate in GNNs (Line 159-160 and 173-174), the authors propose to analyze GNNs using the maximum entropy of hidden features and argue to choose the widths and depth to maximize entropy (Theorem 1, especially Line 183-184).
- While entropy maximization can be done by arbitrarily scaling width and depth of GNNs, in Section 5, the authors provide a rationale for regularizing depth based on a lower-bound of channel capacity (Eq. (16)), which states that increasing model depth leads to a diminishing return in increasing the lower bound.
- Based on arguments in Section 4-5, in Section 6 the authors provide an algorithm for prescribing width and depth of GNNs that maximize entropy under the lower-bound constraint on channel capacity, and an additional constraint that dimensions over depth are non-increasing.
- In Section 7, the authors experiment the algorithm compared to fixed shallow and deep architectures for representative GNNs for 5 semi-supervised learning datasets, measuring model accuracy and entropy, mean, and variance of hidden features.

**Strengths:**

- S1. This paper aims at an important problem of prescribing the model capacity of GNNs before learning.
- S2. The approach from the perspective of information theory taken in this paper is original as far as I am aware.
- S3. The writing is fluent and I was easily able to follow the flow of the paper.

**Weaknesses:**

The main weakness I find for this paper is that the posed problem of choosing widths and depth of GNNs has been not addressed well, both theoretically and empirically. Detailed points are as follows.
- W1. The paper makes a key assumption in Line 159-160 and 173-174 that maximum entropy corresponds to optimal information transmission in GNNs, and I am not sure why or how this assumption can be considered valid (I might be missing something since I am unfamiliar with information theory). This assumption underlies most of the claims of this paper (Theorem 1, especially Line 183-184) and its validity is important.
- W2. I do not see how the results shown in Eq. (16) leads to the conclusion in Line 259-260 that "its negative effects always surpass its merits (typically reflected as model performance degradation). The equation says that adding depth leads to diminishing, but non-negative returns, as the summands are positive for any $w_{l-1}, w_l \geq 2$. This in fact means adding depth should be always beneficial (at least not detrimental).
- W3. The nonlinear programming problem in Eq. (18) which underlies the main algorithm seems to have two issues. First, it does not optimize the depth $L$ of GNNs as far as I understood, and hence does not solve the considered problem of choosing both widths and depth. Second, the termination condition is given by an offset of the constraint parameterized by a new constant $\zeta$, and what this termination condition means in the information-theoretic framework of this paper and how $\zeta$ is chosen is unclear.
- W4. Overall, I am not sure how to interpret the experimental results. For example, in Section 7.2, it is stated that "unlike real communication channels, GNNs might suffer from severe over-fitting due to extra available channel capacity" (Line 355-356), which seems to contradict the theoretical interpretation of GNNs given in this paper based on maximal entropy, as well as the usefulness of the main algorithm. In Section 7.3 the authors state that "C3E variants experience a considerable entropy increase at the first layer... Yet, these characteristics do not necessarily guarantee better performance in the downstream tasks", which raises a similar concern. Given this, I am not sure how the conclusion "GNNs perform similarly to noisy communication channels" in Line 469 can be made. Other detailed points are given in W5-8.
- W5. In Line 347, the authors argue on Figure 1 that "the accuracy curves exhibit continuous U-shaped and V-shaped patterns over the figures", and connect this to scaling laws of language models. I was unable to find the argued patterns in Figure 1, and in fact, it seems that in many cases the accuracy degrades with increased model capacity (e.g., GAT in AmazonComputers).
- W6. In Line 350, the authors say that "C3E variants reach their peak performance within [ϕ0 − log n, ϕ0 − 2log n]". I am not sure what this interval means, because the x-axis of the plots corresponds the control variable ϕ0 − log n.
- W7. In Line 372-374, The authors state that "The massive entropy reduction occurs relatively early in most Deep models, implying diverse features are collapsed into centralized or biased features." The logic behind this argument is unclear. For example, a plausible alternative explanation is that classification quickly finishes in early layers.
- W8. In Section 7.4, I was not sure what specific theoretical results the authors were trying to verify (Line 421) by measuring mean and variance of hidden features. The authors provide Eq. (19) which assumes that all weights and input features come from the standard normal distribution (Theorem 1), and then argue that GNNs with non-zero mean features have learned "biases" in the features (Line 453) by comparing the empirical mean to Eq. (19). A simpler and more plausible explanation could be that these learned GNNs deviate substantially from the assumptions of Theorem 1.
- W9. In Line 482-483, the authors claim that "in an attempt to avoid biased results, other GNNs that do not take the form of Eq. (7) and other aggregation methods are not included." It was unclear to me what kind of biased results the authors were trying to avoid with limited evaluation of model classes. I have a similar concern for Line 1113-1114, where the authors state "For a fair comparison, only deep backbone models and C3E variants have residual connections in the form of Eq. (41)." and it is unclear how this ensures a fair comparison of which models.

**Questions:**

- Q1. In Appendix E.4, how are the hyperparameter configurations for the plain and deep models chosen for each dataset?

---

> ### Author Response · Authors · 2024-11-18
> **Reply to Reviewer e4SK**
>
> Thanks for your review.  We apologize for any misunderstanding - this work (and equation (7)) focuses only on a subset (spectral-based GNNs) of GNNs and we should have made this more clear in our introduction.
>
> Weakness:
>
> 1. First, the neural networks can be seen as an information processing system. Then, based on information theory, its channel capacity phi = max I(Omga(G);G') = max (H(Omga(G)) - H(Omga(G)|G')). Ideally, the neural networks learn the exact mapping between input graph G and processed/target graph G', then H(Omga(G)|G')) = H(G'|Omga(G)))=0. Then, max I(Omga(G);G') = max (H(Omga(G)) - 0) as stated in Eq.(13). Therefore, to reach the optimal information transmission state (i.e., error-free transmission) when modeling a neural network as a communication channel, we have to maximize H(Omga(G)) to ensure that Shannon’s Theorem is satisfied. Shannon’s Theorem and channel capacity formulation are well-established theories in information theory, you can find their mathematical proof in [1].
>
> 2. Eq.(16) does suggest increasing the depth increase the lower bound of the channel capacity, but please refer to Eq.(11), if w_l /d < 1 or the variance of the generalized adjacency matrix \hat{A} at l-th layer < 1/n, then adding more layers will result in the reduction of the model entropy H(Omega) which in turn decreases the channel capacity. Furthermore, in section 7, the empirical results show that given the same phi_0 (the lower bound of the channel capacity), the model with higher model entropy performs better. This reiterates why the maximization of entropy is valid and important.
>
> 3. Please check the reply to Ek4c, questions 4. The programming problem is iteratively solved. It starts with L = 1, and keeps incrementing until the feasible solutions are found or the offset \zeta is reached. The reason for introducing \zeta, is primarily a measurement of channel capacity utilization efficiency. In real-world information processing systems, only a certain proportion of the theoretical channel capacity is used for information propagation, while \zeta typically ranges from 49% to 91%. Therefore, we consider the worst case that satisfies Shannon’s Theorem will take at most 2x of the theoretical maxima. And sorry for assuming readers are familiar with information theory, we will add a description of this consideration in the paper.
>
> 4. First, we are sorry for presenting the notations ambiguously. For your first concern, please refer to Fig. 1. We can observe from the figures that the peak performance only occurs in certain points where log(n) < phi_0 < 2 log(n) (Shannon’s Theorem is surely satisfied when phi_0 >= log(n), and we can reach error-free optimal transmission over the communication channel), this correspond to the point 3. Moreover, we can observe that after reaching the peak performance, where phi_0 is close to 2 log(n) and surpasses it. The average model performance drops, which indicates the presence of over-fitting. As the number of parameters are growing, the possibility of over-fitting increases, therefore resulting in degradation of average performance. For your second concern, please see the reply to Ek4c, question 5. The entropy and channel capacity in this work are the upper bound (maximum entropy) (based on the principle of maximum), not the real-time/exact entropy and channel capacity during the training (we won't know about their exact values until using the model for inference). Therefore, the choice of other hyperparameters (such as activation functions, number of heads in GAT, diffusion time, and expansion steps in GDC) affects the performance of C3E-determined baselines. Carefully tuning these parameters always yields better results compared to baselines, but as replied in Ek4c, question.5, we only change the depth and width. Thus, if you expect consistently better results than baselines, we surely can add them as another table.
>
> 5. We apologize for the narrow figures hindering the presentation of patterns. I mean the U curves and V curves are not the overall curve, but more locally. What I want to express is the change in model size is not monotonically. Starting from a given model, you can either obtain a better one or a worse one by changing the configurations of depth and width.
>
> 6. We are sorry the x-axis labels are a bit confusing, it should be phi_0 \in [log(n), 2log(n)]. As for why 2log(n), please check point 3 (1/0.49999).
>
> 7. Thanks for pointing out the incomprehensive interpretation. We will polish our interpretations based on the resulting figures.

---

> ### Author Response · Authors · 2024-11-18
> **Reply to Reviewer e4SK**
>
> Weakness:
>
> 8. Please check Appendix A.1. One can not estimate the precise statistics (mean and variance are components determining the entropy, see Appendix A.2) before training the model, but we can estimate their upper bound about the entropy, where a continuous random reaches its maximum entropy with a Gaussian distribution. Meanwhile, considering the normalization in the weights, inputs, and following [2,3], we assume N(0, 1^2). Besides, thanks for pointing out the vague interpretations. We will improve our interpretations.
>
> 9. Please check the reply to Ek4c, weakness 1. If the propagation mechanism cannot be represented or collapsed by a matrix, this means that these GNNs (like sampling, pooling, and concatenation over different dimensions) are not spectral-based GNNs, they are spatial-based GNNS. If we do not select the spectral-based GNNs (take the form of Eq.(7)), we cannot estimate depth and width from an analytical form. For the last concern, we acknowledge that the results presented are a bit confusing. The residual connection is optional in the setting, as derived in Appendix B, it helps the model to have higher model entropy (in turn higher channel capacity). We want to show that even with the same depth and all other configurations the same, the appropriate width of each layer matters as well. We attempt to combine the ablation study part into the evaluation part, thus, leading to ambiguity. We will separate the ablation results and add a new subsection to discuss. Besides, we acknowledge that baselines of 3 can be insufficient, there are other spectral-based GNNs such as SGC, APPNP, and SSGC. We will add additional results, but this can take more than 4 weeks.
>
> Questions:
>
> 1. Please see Appendix E (line 1100) for hyperparameter configuration, plain and deep models have the same dropout probability, weight decay, learning rate, etc as the corresponding original works do.
>
>
> [1] A mathematical theory of communication.
>
> [2] Deep-mad: Mathematical architecture design for deep convolutional neural network.
>
> [3] The principles of deep learning theory.

---

> ### Comment · Reviewer_e4SK · 2024-11-27
>
> Thank you for the response, which has helped me better understand the paper. I have some questions upon reading the response.
>
> On W2:
> - Can the authors point to several specific examples in Section 7 which shows that, given the same phi_0, the model with higher model entropy performs better?
> - On Eq. (16) and Eq. (11), is there any concrete reason not to choose w_l sufficiently large, and scale \hat{A} so that its variance becomes > 1/n? The theory of the paper predicts that this would increase the model entropy.
>
> On W4:
> - Why should over-fitting happen for phi_0 >= 2 log(n) specifically?
>
> On W5:
> - I am not sure if I understood the response correctly. What does the fact that model accuracy does not change monotonically tell us? How does it support the main arguments of the paper?
>
> On W7 and W8:
> - What would be the revised interpretations? Their clarity and validity are important and needs to be checked.
>
> On W9:
> - If the theory applies exclusively to spectral-based GNNs, can the authors describe why application to GAT is shown as a main result?

---

> > ### Author Response · Authors · 2024-11-27
> > **Reply to Reviewer e4SK**
> >
> > W2.(1). From Fig. 1, we see, e.g., for Citeseer: GDC (bottom-row second-left) and GCN (top-row second left) with phi_0 - log n = 4 (x-axis), blue line shows GDC has accuracy (0.725) and GCN has accuracy (0.722), and green line also shows GDC has model entropy 12.12 while GCN has model entropy 11. 76 ) - i.e., the higher entropy model has higher accuracy; similarly, in Fig 1 AmazonPhoto GDC and GCN with phi_0 -log n= 2.5, GDC with model entropy 60.03 (compare GCN with model entropy 52.26) performs better.
> >
> > W2.(2). Yes. You can make the variance of \hat{A} sufficiently large and choose a smaller width to realize a similar model entropy. Still, as we are under the principle of maximum entropy, the model entropy is its upper bound. A smaller width will reduce the lower bound \phi_0, which can lead to \phi_0 < log n. This means that though we can have two different models with the same model entropy (their upper bound is the same), whether the \phi_0 >= log n to realize the optimal information transmission state with arbitrarily small error depends on the widths. Therefore, it should be sufficiently large. Also, 'On Over-Squashing in Message Passing Neural Networks: The Impact of Width, Depth, and Topology' this work [1] shows that the width should be sufficiently larger for a graph perspective, though they did not provide any methods for determining the values or ranges for the width or depth.
> >
> > W4. Sorry for the confusion. 2 means 1/0.5 = \zeta, which is the inverse of channel capacity utilization efficiency. Since communication channels cannot fully exploit their theoretical capacity (they only use part of it), so we only care about spectral GNNs with utilization efficiency from 1/\zeta to 1.0. This corresponds to the worst utilization efficiency of 0.5 in real-world communication channels, e.g., Bluetooth networks, wi-fi in congested environments [2]. Thus, models (communication channels) below utilization efficiency of 1/\zeta get over-parameterization (overestimating the complexity of information that should be propagated).
> >
> > W5. The fact that accuracy does not change monotonically suggests that to realize optimal performance, we must choose models of proper size, such that \phi_0 <log n or \phi_0 > \zeta \log n will lead to sub-optimal performance (where phi_0 is a lower bound of \phi determined by width and depth). Therefore, simply choosing an extremely large network is not the best approach We will highlight the solutions of C3E in Figure 1 to differentiate with models using engineering-preferred exponents of 2 (in terms of depth and width).
> >
> > W7. When we say, "diverse features are collapsed into centralized or biased features", we are referring to the fact that entropy is a measurement of uncertainty or information content. E.g., If the learned representation is [0.1, 0.4, 0.7], its entropy is higher than the representation [0.0, 0.0, 0.2]. Then, the more distributed (diverse) is the former, and the more centralized is the latter. Our revised interpretation is: "The large entropy reduction occurs earlier before the classification layer in most Deep models, implying the representation learning process for performing classification is shorter than C3E estimated models. This indicates that simply increasing the depth does not benefit the model to learn more diverse and complex features."
> >
> > W8. As derived in the appendix, under the principle of maximum entropy, the mean and variance of representations determine the model entropy and the lower bound of channel capacity. Our revised interpretation is: "This suggests that simply stacking multiple layers or using shallow models deviates substantially from the assumption in Theorem 1, while C3E estimated models based on the principle of maximum entropy generally fit the assumption."
> >
> > W9. Sorry for the confusion, you are correct that GAT is not a spectral GNN as it does not follow eq (7). We included GAT because we wanted to compare the difference when applying the framework to a spatial GNN. However, we realize that this is causing confusion - we will remove GAT results from Table 1 and we will present GAT results in a separate section in the Appendix.  Furthermore, we will add the results of extra spectral GNNs to the main body.
> >
> > [1] On over-squashing in message passing neural networks: the impact of width, depth, and topology.
> > [2] Spectral efficiency enhancement and energy optimization in 5G networks via stochastic optimization-inspired by modified slime mould algorithm.

---

> ### Comment · Reviewer_e4SK · 2024-11-29
>
> Thank you for the additional response. After reading the responses and other reviews, my revised assessment of the paper is that, while the proposed theory does have interesting points and a potential to be useful, the current paper needs a non-trivial amount of development and polishing before publication.
>
> On the need for improvements, I share most of the concerns with reviewer MwuB and will not repeat them.
>
> On the other hand, as a suggestion for improvement, I recommend the authors to center the paper upon 1-2 specific useful predictions the proposed theory makes, while existing theories (e.g. over-smoothing/squashing analysis, WL hierarchy) do not, and present explicit empirical validation. In this regard, I find it quite interesting that over-fitting seems to (according to the authors) happen around 0.5 of the channel capacity utilization efficiency, which matches the worst utilization efficiency in real-world applications. It would require substantial additional empirical validation, but this type of prediction is not made by prior theories for GNNs, as far as I know. Again, this would require a significant revision of the paper.

---

### Official Review · Reviewer_MwuB · 2024-11-07

**Soundness:** 2
**Presentation:** 1
**Contribution:** 1
**Rating:** 1
**Confidence:** 4

**Summary:**

This paper proposes a take on GNNs by considering the enropy of the node features as a function of depth and width of the network.
The authors present several theorems that characterize the behavior of GNNs when they depth or width increases, and based on that they propose a method called C3E to constrain this entropy.

Experiments and evaluations are provided on a few datasets.

**Strengths:**

The idea and analysis present here are sounds.

**Weaknesses:**

**The author say "This work illustrates that the expressive ability of GNNs
depends on their model capacity, which is directly determined by the widths and depths of the networks." in the introduction, but in the end they never discuss the expressive power of GNNs by the formulation provided here. They also consider rather simple GNNs whose expressiveness is already well established [1,2].

** Poor discussion of related works: while the authors discuss some relevant works like Di Giovanni et al., they do not clearly explain what is the difference between the papers and what is the main benefit of looking at the problem from the proposed perspective.

** Missing discussion of related works: for example [3] also studied the outputs of GNNs and shows similar conclusions, and the well studied phenomenon in GNNs of oversmoothing was shown to yield similar results [4,5]. I thus tend to feel that significant works in the field have been overlooking and there is lack of novelty in this paper, despite its soundness.

** Not all GNNs can be explain by Equation (7). I think that this is a limitation of the method and misses on well known and important architectures like [1,2].

** There are several typos and required improvements to the level of English, for example "Let's" -> "Let us". Another example is a missing word in line 359-360 "Unlike the graph attention mechanism
adaptively weighs the given edges..."

** The derivation in Equation (8) leads to oversmoothing and convergence to the leading eigenvector of the adjacency matrix, which is already a known result in GNNs.  Similar argument holds for lines 200-203. While the discussion is correct, it is not new.

** The experiments are not convincing. According to the authors, they used the public split on Cora/Citesser/Pubmed. However, it is known that GCN obtains better results than reported here. See for example the results reported in [6]. Also, these datasets were shown to suffer from several issues to properly evaluate GNNs, please see [7].

** It is not clear how much time it takes to solve the programming problem? Is it practical for large scale networks?

** Why not show results on additional backbones like GCNII/GIN or even graph transformers like GPS?

** With respect to my previous question, I think that the plots the authors show in Figure 1 are not a general result, because if they use GCNII which does not over smooth, the accuracy will not drop, and it might be that the entropy also doesnt increase as suggested. I would expect a more rigouruous study.

** The authors should show results on additional datasets and benchmarks, such as LRGB, OGB, TUDatasets and others to provide a more compelling paper. As it is now, the results are not convincing and the theoretical derivations while correct, do not offer a new perspective on what we already know.

** File issue: It seems that the figures used in the paper make the rendering of the paper problematic. I suggest that the authors revise them with lighter files.



References:

[1] Weisfeiler and Leman Go Neural: Higher-order Graph Neural Networks

[2] How Powerful are Graph Neural Networks?

[3] Graph neural network outputs are almost surely asymptotically constant

[4] Revisiting Graph Neural Networks: All We Have is Low-Pass Filters

[5] A Note on Over-Smoothing for Graph Neural Networks

[6] Simple and Deep Graph Convolutional Networks

[7] Pitfalls of Graph Neural Network Evaluation

**Questions:**

What do you mean by 'graph mining' in the introduction and abstract of the paper?

---

> ### Author Response · Authors · 2024-11-18
> **Reply to Reviewer MwuB**
>
> Thanks for your review. You have focused on the over-smoothing and expressive power of all message-passing GNNs from a graph theory-based point, which is not the problem we are attempting to address. We apologize for any misunderstanding - this work (and equation (7) in particular) focuses only on a subset (spectral-based GNNs) and we should have made this more clear in our introduction.
>
> Weakness:
> 1. The two papers [1,2] demonstrate the expressive power of spatial-based GNNs grounded on Weisfeiler-Lehman graph isomorphism for graph classification, and we respect these works. Nevertheless, our work differs a lot, it does not focus on the graph properties, but on the depth and the width of the neural network. These works and most existing works do not have direct theoretical frameworks for determining the appropriate depth and width given a graph with a defined layer propagation rule before learning. These two hyperparameters are either selected based on grid search/Bayesian optimization or engineering guidelines, we never know why and the mechanism behind them. For your last concern, we acknowledge that the baselines might be insufficient. Again, this work is not proposing a new generation of message-passing GNNs, we did not propose new propagation rules, and we only care about determining the optimal depth and width before training.  As for why choosing GNNs taking the form of Eq.(7), other spatial-based GNNs that you think are not simple, such as GIN, GraphSage, GPS, etc, all have non-linear operations, which are impractical and impossible to directly transform into closed form linear algebra. E.g., sampling, pooling, and concatenation over different dimensions. If the utilization of connectivity information in the propagation rule can be collapsed or represented by a matrix (often spectral-based GNNs), then it in general fits Eq.(7), and it can be explained with linear algebra. To my best knowledge, there is no such work that can unify all GNNs with these discrete and non-linear operators in the propagation rules and represent them in a basic matrix operation (and also one of the reasons we consider spectral-based GNNs).
> 2. Thank you for your suggestions. We acknowledge that the discussion of related work and emphasis on the differences and benefits of the proposed method might not be sufficient. We will add more descriptions of spatial-based GNNs and spectral-based GNNs, and how spectral-based GNNs benefit from using C3E.
> 3. As responded in point 1., the suggested works are mainly focusing on tackling over-smoothing from various perspectives. But, our work never claims this. We are presenting a theoretical framework to determine the optimal depth and width of GNNs, this involves considering properties of input graphs, propagation rules, depth, and width. We respect and have read these works, existing works are either concentrated on a graph theory perspective or solely exploring the depth (with width 2^x, 16, 32, 64, 128, ...). There is no such framework or theory to describe why the width is 2^x or that specific depth. We attempt to uncover the mask of good engineering outcomes, which are searched given the interval or manually selected based on non-trivial experience.
> 4. Please check point 1., and we apologize for not clearly stating our target GNNs, spectral-based GNNs.
> 5. Thank you for pointing out these mistakes, we will correct any typos in this work.
> 6. Yes, Eq.(8) is a well-known result in the field, but we never claim this is our original derivation or conclusion. As described in point1., we focus on providing a theoretical framework for determining the optimal depth and width of spectral-based GNNs before training. How to tackle over-smoothing or over-squashing is never the purpose of this work.
> 7. We run all models independently 10 times and we provide the mean value, we should not just copy the results directly from the work. The differences between the real implementation and the reported performance are not statistically significant, showing that we did not deliberately suppress the original works for better comparison, we show what the experiment exhibits. In terms of datasets, we are aware of potential problems, thus we have AmazonPhotos and AmazonComputers. We will experiment on OGB and LRGB in terms of node classification and add the corresponding results. However, this will take more than 4 weeks for us to complete additional results.
> 8. The solving process is easily done within 1min50s on an average CPU, with no GPU required. Since the constraints are log-based, it is practically possible for large-scale graphs. We can add a table to show the time consumption statistics of solving the problem in different scenarios.
> 9. Please check point 1 and point 10.

---

> ### Author Response · Authors · 2024-11-18
> **Reply to Reviewer MwuB**
>
> Weakness:
>
> 10. GCNII shows robust results even when stacking deeper and provides a way to alleviate over-smoothing, but our work does not focus on tackling over-smoothing. GCNII has a propagation rule of multiplication between a weight learned representation plus weighted initial residual and a weighted identity matrix plus weighted learnable matrix. It analyzes spectral-based GNNs taking the form of Eq. (7), but the proposed propagation rule cannot be represented in a generalized form of Eq.(7), which deviates from the spectral filtering that spectral-based GNNs do. The weighting coefficients are critical for GCNII, it degenerates to Eq.(7) if the weight coefficients are 1. In terms of Fig.1, the model entropy is the theoretical maxima, adding more layers or increasing the widths > \sigma^2(A_l) will increase the model entropy, see Eq.(11), i.e., the maximum information that can be retained. The premise is the principle of maximum entropy, thereby, the entropy in this work we considered is the known upper bound rather than the exact value. As one cannot estimate the precise model entropy before learning/training.
>
> 11. Please check point 7. We can add comparisons between spectral-based GNNs and spatial-based GNNs such as Graph Transformers, GIN, and GraphSage. But, if we go beyond learning for nodes, i.e., to graphs, then this is the problem that another paper should consider. We have to trade off between graphs with different sizes and dimensions of features, leading to a complex multi-level optimization problem to generate the optimal width and depth for all graphs. This will be the future work.
>
> 12. Thank you for the suggestion. We will polish and improve the figures for better presentation and comprehension.
>
> Questions:
>
> 1. This means the representation learning from the graph.

---

> > ### Comment · Reviewer_MwuB · 2024-11-24
> >
> > I thank the authors for the responses. I will maintain my score at this point because I think that the paper at its current state is not sufficient for publication at ICLR. I certainly think that more experiments are required to evaluate this method, to understand how the proposed method can be used on many different types of tasks and inputs.
> > Also, I think that it is needed to know how does the method related to known problems like oversmoothing and oversquashing - these are definitely related to the width and depth of the network, see [1] below for example.
> > Finally, I still think that the paper needs to be substantially revised to understand what is new here and what relies on previous studies, as discussed in my review and now acknowledged by the authors.
> >
> > As a general framework, this might be an interesting study. But, it needs to be further developed.
> >
> > [1] On Over-Squashing in Message Passing Neural Networks: The Impact of Width, Depth, and Topology

---

### Official Review · Reviewer_iUz9 · 2024-11-10

**Soundness:** 2
**Presentation:** 2
**Contribution:** 2
**Rating:** 5
**Confidence:** 1

**Summary:**

Message-passing graph neural networks are shown to function like noisy communication channels, with optimal performance achieved when Shannon's theorem balances entropy and channel capacity. The study highlights that trainable matrix widths must be large enough to maintain channel capacity, which diminishes with increasing network depth, and validates these findings through experiments on five node classification datasets.

**Strengths:**

I am not an expert in this field and struggled to fully understand the derivations of the theorems, so I am unable to thoroughly evaluate the strengths and weaknesses of this paper. I apologize and will defer to the insights provided by other reviewers.

**Weaknesses:**

I am not an expert in this field and struggled to fully understand the derivations of the theorems, so I am unable to thoroughly evaluate the strengths and weaknesses of this paper. I apologize and will defer to the insights provided by other reviewers.

**Questions:**

I am not an expert in this field and struggled to fully understand the derivations of the theorems, so I am unable to thoroughly evaluate the strengths and weaknesses of this paper. I apologize and will defer to the insights provided by other reviewers.

---

### Official Review · Reviewer_Ek4c · 2024-11-11

**Soundness:** 1
**Presentation:** 2
**Contribution:** 1
**Rating:** 3
**Confidence:** 4

**Summary:**

The authors argue that graph neural networks behave similarly to noisy communication channels. Based on information theory, they derive a nonlinear programming that maximizes the entropy of a GNN, thereby finding its optimal transmission state. By solving this optimization problem, they aim to find optimal network architecture parameters, i.e. the number of layers (network depth) and the size of the weight matrices for each layer (network width). In semi-supervised node classification experiments, they compare the optimized network architectures of some well-known GNNs to commonly used architecture configurations.

**Strengths:**

1) The authors have a good motivation for their work, which is solving the problem of finding optimal network widths and depths.
2) The necessary theoretical background to understand their approach (e.g. information theory, entropy, communication channels) is well explained.

**Weaknesses:**

Major (Content):

1) The assumption that Cl = A (see line 152) is valid for GCN only. The proposed framework is based on this assumption. However, the authors use their framework also for GAT and GDC (see Sec. 7 “Experiments”).

2) The proposed optimization problem seems to lead to a very large network width (> 1000, see App. E). This leads to a significantly larger number of parameters for the C3E-models compared to the “Plain-“ and “Deep-“ models, which have network widths of at most 64. In fact, the total number of trainable parameters for the C3E-models is larger than 1.000.000 compared to ~20.000 for the baselines, which seems to be an unfair comparison.

3) The optimization problem in Eq. 18 does not consider computational efficiency. At some point, increasing network width and depth may not be computationally feasible anymore or the resulting performance improvement may not be worth the computational expense.

Major (Formal):

a) Fig. 1-4: The subfigures are way too small and it is hard to read the axis labeling. Maybe it would be better to show these results only for one or two of the datasets and shift the other results to the Appendix.

Minor:

  1) Although the introduction contains a good motivation (how to balance receptive field and over-smoothing by finding optimal GNN width and depth), the introduction lacks information on the proposed approach. There is only one sentence from line 69-71, in which the authors mention (very briefly) what they are doing in the paper.

2) The section “other related work” lacks some clear differentiation of the author’s work from the literature.

3) Eq. 1: Integral domain is missing.

4) Eq. 4: Please use a different index in the sums in denominator to distinguish it from the index of the outer sum.

5) Sec. 4 introduces some more information theory basics in the second paragraph. This could be shifted into Sec. 3.

6) Eq. 19 is identical to Eq. 9 and 10. It is reader-friendly to recall these equations, but using different numbers might be confusing

**Questions:**

1)  Eq. 8: I do not quite understand, why the activation function was removed, which is the only difference between Eq. 8 and Eq. 7. It may not affect the maximum entropy, as the authors argue, but this is not a sufficient reason to just delete it from Eq. 7. Could you please explain this in more detail?

2)  Eq. 11: As far as I understood, Eq. 10 is used for the approximation in Eq. 11. However, I do not understand, where the second term in the second line of Eq. 11 comes from.

3)  Eq. 18: What do you mean by L=2 below the sum?

4) How do we know that there is always a solution to the optimization problem in Eq. 18?

5) The main idea of the paper is to find a GNN’s optimal transmission state by maximizing its entropy, which is defined in Eq. 11. However, in the Discussion the authors state that “… reaching the optimal information transmission state does not necessarily guarantee improvements in model performance.” This leads to the question of why the proposed optimization is necessary at all. Could you please explain what you meant by that in more detail?

---

> ### Author Response · Authors · 2024-11-18
> **Reply to Reviewer Ek4c**
>
> Thanks for your review. We realize that we have been ambiguous in our description of "message-passing GNNs" to which this work refers. We said "message passing GNNs, we should have been more explicit and said spectral-based GNNs, where C_l is not dependent on latent representation. But, other spatial-based GNNs do, such as GIN, GraphSage, and GPS (Graph Transformers), the propagation mechanisms (aggregation, pooling, sampling, concatenation) cannot be collapsed into a matrix C_l and are latent feature dependent. This makes spatial-based GNNs impractical to analytically studied in closed form.
>
> Weakness (major):
> 1. As line 152 stated, C_l denotes the matrix represents the propagation mechanism, we set Cl = A = D-½ A D-½ as an example to illustrate. The formal form is presented in Eq.(28) in the Appendix, and also in Eq.(18), it is an example of usage, in practice C_l varies according to different spectral propagation mechanisms. As previously mentioned, spatial-based GNNs like GIN, GraphSage, and GPS are not included due to their incorporation of concatenation over different dimensions, sampling, Min/Max pooling, and aggregation of propagation rules. These cannot be represented in a basic linear algebra with an analytical form.
> 2. C3E just determines the depth and width with no modifications to the propagation rules of baselines, which reiterates the core of this work is to find the optimal depth and width before learning. Except for using estimated depth and width, C3E variants are not different from the original baselines, this is a fair comparison. The depth and width of original baselines are manually set or searched over given intervals, while C3Es' come from solving a programming problem.
> 3. The programming problem in Eq.(18) can be easily solved with a CPU within an average time consumption of 1min50s for the feasible solutions. If the users are concerned about the computational resources, they can easily implement a termination rule w.r.t Flops, or the number of parameters over the solver.  All models with estimated width and depth can be run on the free version of Colab with a 12GB RAM T4 GPU.
>
> Major (Formal):
> a) Thank you for your suggestion. We will further polish the figures to present the two most representative figures and move others to the appendix.
>
> Minor:
> 1. Thank you for your advice. We will add more descriptions of the proposed work and how it relates to Shannon's Theorem and the principle of maximum entropy.
> 2. Thank you for your advice. We will add more comparisons between related works and the proposed method to emphasize our contribution - to estimate the width and depth of spectral-based GNNs in their optimal transmission state before learning.
> 5. Thank you for your suggestion. We will shift more information theory basics in early sections to help readers not be misled.
> 3 - 4 and 6. Thank you for your suggestion. We will re-write any ambiguous notations to avoid misunderstandings.
>
> Questions:
> 1. First, as stated in line 162 to line 167. The latent representation H_l is a matrix of shape [n, w_l], the target distribution is unknown in advance, therefore its maximum entropy can be calculated with Eq.(3), only a function of dimensionality. The activation function is applied to the elements in H_l, which does not contribute to the maximum entropy. The premise is the principle of maximum entropy, thereby, the entropy in this work we considered is the known upper bound rather than the exact value influenced by non-linear transformations. E.g., Consider a [2, 2] matrix its maximum entropy is ln(2x2) if 4 elements are different from each other, and ReLU truncates negative values and reduces the entropy to ¾ ln(3) + ¼ ln(4). We cannot precisely estimate the entropy of H_l. but we can know its known upper bound.
> 2. Please see line 761 to line 768 and Eq.(27) in line 746. As indicated, a linear layer reshapes the learned representation, then H_L = H_{L-1}W_L, further leading to w_L \sigma(U_{L}), and the linear layer does not have C_l (i.e., A), then w_L \sigma(H_{L}). Therefore, ½ ln(w_L) is presented in Eq.(11).
> 3. We are sorry this notation in the formula is a bit confusing. L = 2 in Eq.(18) indicates that the solving is an iterative process (the 1 in it is the linear reshaping layer with no activation) that starts with 1 message-passing layer, once the solution of 1 layer is generated and then it goes to the 2 message-passing layer case and so forth. Therefore, the configuration of depth and its corresponding width is found.

---

> > ### Author Response · Authors · 2024-11-18
> > **Reply to Reviewer Ek4c**
> >
> > Questions:
> >
> > 4. As SLSQP is a quasi-Newton method applied to a Lagrange function consisting of the loss function and equality- and inequality constraints. In this work, Eq.(18) is most logarithms-based on w_l (we round the decimal digits of estimated w_l in practice), which are continuous and concave functions on (0,∞), and by selecting w_l large enough, we can always ensure that the left-hand side exceeds ln(n), thus satisfying the constraint. Weierstrass' Theorem states that if a continuous function is defined on a compact (closed and bounded) feasible set, then it attains its maximum and minimum on that set. A. The feasible set is bounded below (since w_l>=2) and can be considered bounded above if we introduce upper bounds (which might be necessary in practical applications to prevent variables from going to infinity). B. The objective function is continuous in w_l over the feasible set. Therefore, there always exists a solution.
> > 5. We are sorry that the presentation of results is ambiguous, we tried to incorporate an ablation study in it. As the response to Weakness.2 describes, C3E is a theoretical framework to determine the depth and width for spectral-based GNNs that take the form of Eq.(7). We do not change the propagation rule of the baselines, but their depth and width. The entropy and channel capacity in this work are obtained by considering the maximum entropy case (based on the principle of maximum principle), not the real-time entropy and channel capacity during the training. Therefore, the choice of other hyperparameters affects the performance of C3E-determined baselines. We had better results of C3E-determined baselines (always outperform the baselines) with well-tuned hyperparameters except for width and depth. But, from the ablation study, this can result in more than two variables (depth and width) changing in the experiment. Therefore, we made this conclusion, which originally means that increasing the parameters in a principled way provides the theoretical ability to reach optimal transmission state (error-free transmission), but other hyperparameters (hyperparameters in different propagation rules) influence the practical realization of model performance. We will show the consistently improved performances in the evaluation part and separately create an ablation study part to show the difference between the well-tuned hyperparameters scenarios and the un-tuned hyperparameters scenarios.

---

### Note · Authors · 2025-05-06

I have read and agree with the venue's withdrawal policy on behalf of myself and my co-authors.

---

### Meta-Review · Area_Chair_NHzz · 2024-12-20

**Metareview:**

**(a) Scientific Claims and Findings:**
Message-passing Graph Neural Networks (GNNs) are interpreted as noisy communication channels. By applying principles from information theory, specifically Shannon's theorem and the principle of maximum entropy, the authors suggest that the optimal information transmission in GNNs is achieved when certain conditions related to entropy and channel capacity are met. They argue that the width of trainable matrices should be sufficiently large to prevent a reduction in channel capacity and that increasing network depth leads to diminishing returns in channel capacity enhancement.

**(b) Strengths:**
* Novel Perspective: The paper introduces an innovative viewpoint by modeling GNNs as noisy communication channels, bridging concepts from information theory and graph neural networks.
* Theoretical Insights: Utilizing Shannon's theorem and the principle of maximum entropy provides a theoretical framework to understand information flow within GNNs, potentially guiding architectural design choices.
* Empirical Validation: The study includes experiments on five public semi-supervised node classification datasets, offering empirical support for the proposed theoretical claims.

**(c) Weaknesses:**
* Theoretical Rigor: While the paper presents an intriguing analogy, the theoretical development lacks formal proofs and detailed derivations, which are essential to substantiate the claims rigorously.
* Empirical Evaluation Depth: The experimental results, though present, are not exhaustive. The paper would benefit from a broader range of experiments, including comparisons with state-of-the-art models and evaluations on additional tasks beyond node classification.
* Practical Implications: The discussion on how the theoretical insights translate into practical guidelines for designing GNN architectures is limited, reducing the immediate applicability of the findings.
* Clarity and Presentation: Certain sections of the paper are dense and lack clarity, making it challenging for readers to fully grasp the proposed concepts and their implications.

**(d) Reasons for Rejection:**
After a thorough evaluation, the decision to reject the paper is based on the following considerations:
1. Insufficient Theoretical Foundation: The paper's theoretical claims are not adequately supported by formal proofs or detailed derivations, undermining the robustness of the proposed analogy between GNNs and noisy communication channels.
2. Limited Empirical Support: The experimental validation lacks depth and breadth, with insufficient comparisons to existing state-of-the-art methods and a narrow focus on node classification tasks, which does not comprehensively demonstrate the efficacy of the proposed approach.
3. Unclear Practical Applications: The manuscript does not effectively bridge the gap between theoretical insights and practical applications, offering limited guidance on how the findings can inform the design and optimization of GNN architectures in real-world scenarios.
4. Presentation Issues: The clarity and organization of the paper are suboptimal, hindering the reader's ability to understand and assess the contributions fully.
To enhance the paper for future submissions, the authors should focus on providing rigorous theoretical proofs, expanding empirical evaluations to include a wider array of tasks and comparisons, elucidating the practical implications of their findings, and improving the overall clarity and presentation of the manuscript.

**Additional Comments On Reviewer Discussion:**

The reviewers agree that the current paper is not ready for publication and made many suggestions to improve the presentation and analysis.

In general, the authors did not address many of the raised concerns but promised future changes. For instance, the authors promised in response to Reviewer Ek4c to show the consistently improved performances in the evaluation part and separately create an ablation study part to show the difference between the well-tuned hyperparameters scenarios and the un-tuned hyperparameters scenarios. Reviewer MwuB found the experiments unconvincing because the baselines can reach higher accuracies and requested additional experiments on LRGB, OGB, and TUDatasets. Reviewer MwuB also suggested that the relationship to oversmoothing and oversquashing should be analysed to understand the merit of the suggested framework.

---

### Decision · Program_Chairs · 2025-01-22

Reject